

# Sponges of Western Mediterranean seamounts: new genera, new species and new records

Julio A. Díaz[1,2], Sergio Ramírez-Amaro[1,3] and Francesc Ordines[1]

[1] Instituto Español de Oceanografía, Centre Oceanogràfic de Balears, España, Palma, Spain
[2] Interdisciplinary Ecology Group, Biology Department, Universitat de Les Illes Balears, Palma, Spain
[3] Laboratori de Genètica, Biology Department, Universitat de Les Illes Balears, Palma, Spain

Corresponding author
Julio A. Díaz, julio.diaz@ieo.es

## ABSTRACT

**Background:** The seamounts Ses Olives (SO), Ausias March (AM) and Emile Baudot (EB) at the Mallorca Channel (Balearic Islands, western Mediterranean), are poorly explored areas containing rich and singular sponge communities. Previous works have shown a large heterogeneity of habitats, including rhodolith beds, rocky, gravel and sandy bottoms and steeped slopes. This diversity of habitats provides a great opportunity for improving the knowledge of the sponges from Mediterranean seamounts.

**Methods:** Sponges were collected during several surveys carried out by the Balearic Center of the Spanish Institute of Oceanography at the Mallorca Channel seamounts. Samples were obtained using a beam-trawl, rock dredge and remote operated vehicle. Additional samples were obtained from fishing grounds of the Balearic Islands continental shelf, using the sampling device GOC-73. Sponges were identified through the analysis of morphological and molecular characters.

**Results:** A total of 60 specimens were analyzed, from which we identified a total of 19 species. Three species and one genus are new to science: *Foraminospongia balearica* **gen. nov. sp. nov.**, *Foraminospongia minuta* **gen. nov. sp. nov.** and *Paratimea massutii* **sp. nov.** *Heteroxya* cf. *beauforti* represents the first record of the genus *Heteroxya* in the Mediterranean Sea. Additionally, this is the second report of *Axinella spatula* and *Haliclona* (*Soestella*) *fimbriata* since their description. Moreover, the species *Petrosia* (*Petrosia*) *raphida*, *Calyx* cf. *tufa* and *Lanuginella pupa* are reported for the first time in the Mediterranean Sea. *Petrosia* (*Strongylophora*) *vansoesti* is reported here for the first time in the western Mediterranean Sea. *Haliclona* (*S.*) *fimbriata* is reported here for the first time in the north-western Mediterranean Sea. *Hemiasterella elongata* is reported here for the second time in the Mediterranean Sea. The species *Melonanchora emphysema*, *Rhabdobaris implicata*, *Polymastia polytylota*, *Dragmatella aberrans*, *Phakellia ventilabrum* and *Pseudotrachya hystrix* are reported for first time off Balearic Islands. Following the Sponge Barcoding project goals, we have sequenced the Cytochrome Oxidase subunit I (*COI*) and the *28S* ribosomal fragment (C1–D2 domains) for *Foraminospongia balearica* **sp. nov.**, *Foraminospongia minuta* **sp. nov.**, *H.* cf. *beauforti* and *C.* cf. *tufa*, and the *COI* for *Paratimea massuti* sp. nov. We also provide a phylogenetic analysis to discern the systematic location of *Foraminospongia* **gen. nov.**, which, in accordance to skeletal complement, is placed in the Hymerhabdiidae

family. A brief biogeographical discussion is provided for all these species, with emphasis on the sponge singularity of SO, AM and the EB seamounts and the implications for their future protection.

## INTRODUCTION

Seamounts are structures of high ecological and biological interest (*Carvalho et al., 2020*; *Morato et al., 2013*; *Rogers, 2018*), which provide excellent habitat for a rich communities of filter-feeding animals, such as corals, crinoids and sponges (*Samadi, Schlacher & De Forges, 2007*). These organisms are favored by enhanced currents, scarcity of fine sediment, accidented topography and predominance of hard substrata, features that characterize seamounts (*White & Mohn, 2004*). Sponges are ubiquitous on seamounts, where they tend to form dense and diverse aggregations that provide habitat and refuge to other animals like crustaceans, mollusks and fishes (*Samadi, Schlacher & De Forges, 2007*). Also, they are involved in benthic-pelagic coupling and recycling of nutrients, both processes of utmost importance in oligotrophic areas like the Mediterranean Sea, where they may contribute to the maintenance of higher trophic levels (*De Goeij et al., 2013*).

Despite their importance, very little is known about sponges of the Mediterranean seamounts, which is in contrast to the vast number of studies on sponge taxonomy available in other domains like the continental shelf or the submarine canyons (*e.g. Vacelet, 1961*, *1969*; *Pulitzer-Finali, Hadromerida & Poecilosclerida, 1978*; *Pulitzer-Finali, 1983*; *Boury-Esnault, Pansini & Uriz, 1994*; *Pansini, Manconi & Pronzato, 2011*; *Bertolino et al., 2015*; *Longo et al., 2018*; *Manconi et al., 2019*; *Enrichetti et al., 2020*). However, in recent years the increase in the use of Remote Operated Vehicles (ROV) has facilitated the access and study of seamounts. Currently, information on sponges is available from the Erathostenes seamount in the Levantine Sea (*Galil & Zibrowius, 1998*), the Vercelli seamount in the northern Tyrrhenian Sea (*Bo et al., 2011*), the Ulisse and Penelope seamounts in the Ligurian Sea (*Bo et al., 2020*), the Avempace, Alboran Ridge, Seco de los Olivos and Cabliers seamounts in the Alboran Sea (*Boury-Esnault, Pansini & Uriz, 1994*; *Pardo et al., 2011*; *Sitjà & Maldonado, 2014*; *De la Torriente et al., 2018*; *Corbera et al., 2019*), and the Stone Sponge, Ses Olives, Ausias March and Emile Baudot seamounts in the Balearic Sea (*OCEANA, 2011*; *Aguilar et al., 2011*; *Maldonado et al., 2015*). However, most of these works adress the sponges at a community level, focusing on a general habitat characterization. Nonetheless, the studies adressing taxonomy have revealed that the Mediterranean seamounts are habitats for rare, poorly-known, or new species. For example, *Aguilar et al. (2011)* reported the carnivorous sponge *Lycopodina hypogea* (*Vacelet & Boury-Esnault, 1996*) at the Ausias March seamount, representing the first sighting of this species outside littoral caves. A singular reef formed by the Lithistid

*Leiodermatium pfeifferae* (*Carter, 1873*) was recorded at the Stone Sponge seamount, being the first report of this species in the Mediterranean Sea (*Maldonado et al., 2015*).

Determining which species are present on a given seamount, and hence the seamount's biodiversity is a first step towards the development of management plans to protect these habitats. It is also crucial to understand seamounts' biocenosis, their structure and dynamics, how they can be affected by human disturbances, and to monitor potential biological invasions and long-term community changes (*Clark et al., 2012*; *Danovaro et al., 2020*).

Sponges are problematic as they are difficult to identify, which may lead to incorrect or underestimated biodiversity values. The use of molecular markers, a powerful tool to help in sponge identification, has shown that this group is much more specious than previously thought, and cryptic species are very common (*Cárdenas, Perez & Boury-Esnault, 2012*). Thus, detailed morphological descriptions supported by a complete genetic database are crucial for future studies.

The objective of this work was to improve the taxonomic knowledge on the sponges at three seamounts of the Mallorca Channel in the Balearic Islands: Ses Olives, Ausias March and Emile Baudot. Currently, these seamounts are being assessed for inclusion in the Natura 2000 network, under the scope of the LIFE IP INTEMARES project. One of the goals of this project is to improve the scientific knowledge of areas of ecological interest that harbor rich, vulnerable and protected habitats and species, which is necessary knowledge for the development of management plans. High abundance and diversity of invertebrates were observed during several surveys carried out in 2018, 2019, and 2020 at these seamounts, highlighting sponges as the dominant group. In the present paper we provide detailed descriptions of 18 demosponges and one hexactinellida, including a new genus and three new species, together with new descriptions and records of poorly-known taxa. For the new and dubious species, the sequences of two most used barcoding genes, the mitochondrial Cytochrome Oxidase subunit I (*COI*) and the nuclear *28S* ribosomal fragment (C1-D2 domains), are also provided.

## MATERIALS AND METHODS

### Study area

The Mallorca Channel is located in the Balearic Promontory (western Mediterranean Sea), between the islands of Mallorca and Ibiza. The area harbors three seamounts: Ses Olives (SO; 1°58′58.8″N, 38°57′36″E) and Ausias March (AM; 1°49′4.8″N, 38°44′49.2″E) located east of Ibiza and Formentera islands, and Emile Baudot (EB; 2°30′0″N, 38°43′55.2″E) located south of Mallorca and east of Ibiza-Formentera (Fig. 1). The seamounts SO, AM and EB are 375, 264 and 600 m high, respectively and 10 to 17 km long, with tabular summits elongated in NE-SW trends and located at 225–290, 86–115 and 94–150 m depth, respectively. SO and AM are of orogenic origin, emerging from depths around 900 and 600 m in their eastern sides and being separated from Ibiza and Formentera islands by depths around 600 and 400 m. By contrast, EB is a guyot of volcanic origin, which in its western side emerges from a plain around 900 m deep, with numerous fields of pockmark type depressions, located between SO and AM. At the eastern side

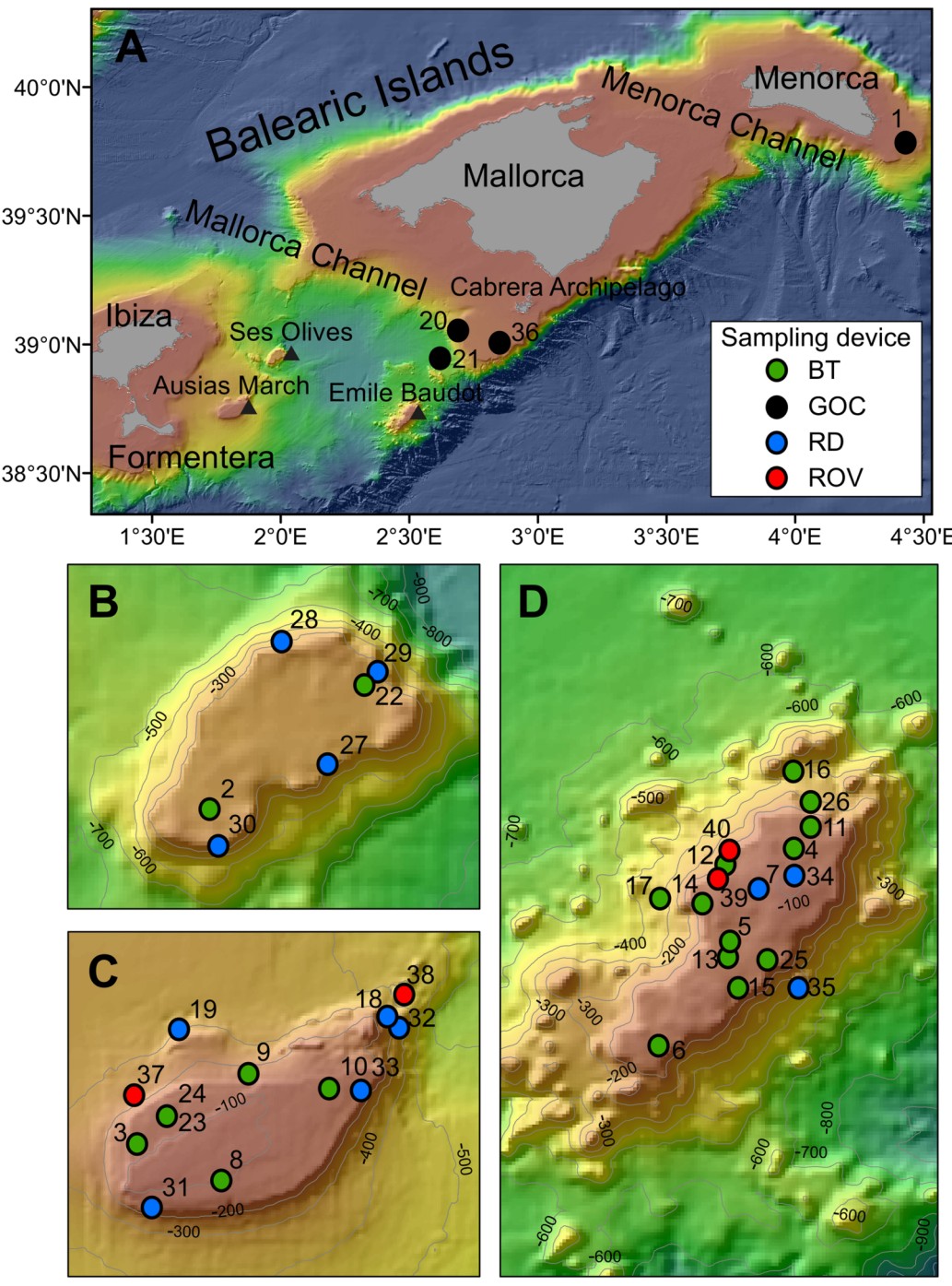

**Figure 1 Map of the studied area showing the location of the sampling stations of beam trawl (BT), bottom trawl type GOC73 (GOC), rock dredge (RD) and remote operated vehicle (ROV). The characteristics of these sampling stations are shown in Table 1.** (A) General view of the Balearic Islands. (B) Detail of Ses Olives. (C) Detail of Ausias March. (D) Detail of Emile Baudot.

**Table 1  Details of the sampling stations.**

| $R_{survey}$ | $R_{study}$ | Year | Sampling device | Depth (m) | Coordinates | | Area | Seabed characteristics |
|---|---|---|---|---|---|---|---|---|
| | | | | | Initial | Final | | |
| 206 | 1 | 2017 | GOC-73 | 135 | 39°47′37.2″N 4°26′15.4″E | 39°47′37.2″N 4°26′15.4″E | E Me | Fishing ground, sedimentary bottom |
| 20 | 2 | 2018 | BT | 275 | 38°56′6″N 1°57′58.3″E | 38°56′6″N 1°57′43.9″E | SO | Detrital bed of muddy sand |
| 22 | 3 | 2018 | BT | 105 | 38°44′30.5″N 1°46′5.9″E | 38°44′30.5″N 1°45′53.3″E | AM | Rhodolith bed with invertebrates |
| 51 | 4 | 2018 | BT | 128 | 38°44′53.9″N 2°30′41.4″E | 38°44′58.9″N 2°30′54.7″E | EB | Coarse sand with dead rhodoliths |
| 60 | 5 | 2018 | BT | 138 | 38°43′13.1″N 2°29′29.4″E | 38°43′5.5″N 2°29′20.4″E | EB | Coastal detrital with sand |
| 66 | 6 | 2018 | BT | 146 | 38°41′13.9″N 2°28′11.3″E | 38°41′7.1″N 2°28′1.9″E | EB | Coastal detrital with sand and small dead rhodoliths |
| 52 | 7 | 2018 | RD | 109 | 38°44′13.2″N 2°30′3.6″E | 38°44′12.5″N 2°30′12″E | EB | Rhodolith bed |
| 50 | 8 | 2019 | BT | 102 | 38°43′33.6″N 1°48′12.6″E | 38°43′34.7″N 1°48′23.4″E | AM | Rhodolith bed with invertebrates |
| 99 | 9 | 2019 | BT | 131 | 38°46′20″N 1°48′54.7″E | 38°46′29.3″N 1°49′36.1″E | AM | Coastal detrital with sand and sponges |
| 104 | 10 | 2019 | BT | 118 | 38°45′57.6″N 1°51′2.5″E | 38°46′4.8″N 1°51′8″E | AM | Coastal detrital |
| 124 | 11 | 2019 | BT | 152 | 38°45′19.1″N 2°31′0.5″E | 38°45′20.9″N 2°31′8.4″E | EB | Detrital border |
| 135 | 12 | 2019 | BT | 169 | 38°44′42.7″N 2°29′25.8″E | 38°44′21.2″N 2°29′15.8″E | EB | Detrital border with sand |
| 136 | 13 | 2019 | BT | 147 | 38°44′42.7″N 2°29′25.8″E | 38°43′13.1″N 2°29′21.5″E | EB | Detrital border with gross black sand |
| 166 | 14 | 2019 | BT | 433 | 38°44′3.1″N 2°28′12.7″E | 38°43′44.4″N 2°28′1.2″E | EB | Detrital mud |
| 167 | 15 | 2019 | BT | 151 | 38°42′21.6″N 2°29′37.3″E | 38°42′12.6″N 2°29′29.4″E | EB | Detrital border with sand |
| 175 | 16 | 2019 | BT | 410 | 38°46′21″N 2°30′44.3″E | 38°46′31.1″N 2°31′5.9″E | EB | Detrital mud |
| 177 | 17 | 2019 | BT | 156 | 38°43′57.7″N 2°28′54.1″E | 38°43′47″N 2°28′53.4″E | EB | Detrital border with sand |
| 95 | 18 | 2019 | RD | 275–220 | 38°47.8′0″N 1°52.6′0″E | 38°47.7′0″N 1°52.4′0″E | AM | Rocky slope |
| 103 | 19 | 2019 | RD | 302–231 | 38°47.4′0″N 1°47.2′0″E | 38°47.3′0″N 1°47.2′0″E | AM | Rocky slope |
| 224 | 20 | 2019 | GOC-73 | 252 | 39°3′3.6″N 2°42′2.9″E | 39°5′15.7″N 2°42′13.3″E | SW Ca | Fishing ground, sedimentary bottom |
| 225 | 21 | 2019 | GOC-73 | 754 | 38°57′11.5″N 2°37′54.1″E | 39°0′2.9″N 2°38′33″E | SW Ca | Fishing ground, bathyal mud |
| 1 | 22 | 2020 | BT | 289 | 38°58′0.5″N 2°0′22.7″E | 38°58′14.9″N 2°0′0″E | SO | Detrital with encrusting sponges and small crustaceans |

(Continued)

| $R_{survey}$ | $R_{study}$ | Year | Sampling device | Depth (m) | Coordinates Initial | Final | Area | Seabed characteristics |
|---|---|---|---|---|---|---|---|---|
| 17 | 23 | 2020 | BT | 113 | 38°45′15.5″N 1°46′53.4″E | 38°45′4.7″N 1°46′36.1″E | AM | Rhodolith bed with invertebrates |
| 18 | 24 | 2020 | BT | 114 | 38°45′15.5″N 1°46′53.4″E | 38°45′16.2″N 1°46′54.1″E | AM | Rhodolith bed with invertebrates |
| 45 | 25 | 2020 | BT | 147 | 38°42′51.8″N 2°30′13.7″E | 38°42′28.1″N 2°29′24″E | EB | Coarse sand and gravel with crustaceans and sponges |
| 52 | 26 | 2020 | BT | 320 | 38°45′47.5″N 2°31′0.5″E | 38°45′56.9″N 2°30′37.1″E | EB | Organogenic sediments, shells rests and gravel with sponges |
| 3 | 27 | 2020 | RD | 288–318 | 38°56′4.7″N 1°59′48.1″E | 38°56′44.5″N 1°59′46.3″E | SO | Rocks and rests of fossil Ostreids |
| 7 | 28 | 2020 | RD | 325–255 | 38°58′41.9″N 1°59′2.4″E | 38°58′33.6″N 1°59′8.5″E | SO | Rocks, rests of fossil Ostreids and fossil corals |
| 8 | 29 | 2020 | RD | 315–295 | 38°58′11.3″N 2°0′30.6″E | 38°58′12″N 2°0′25.2″E | SO | Rocks and rests of fossil Ostreids |
| 14 | 30 | 2020 | RD | 325–270 | 38°55′33.6″N 1°58′5.6″E | 38°55′45.1″N 1°58′1.2″E | SO | Mud, rocks and fossil Ostreids |
| 20 | 31 | 2020 | RD | 104–138 | 38°42′51.1″N 1°46′28.2″E | 38°43′14.5″N 1°46′27.5″E | AM | Rhodolith bed with sponges |
| 27 | 32 | 2020 | RD | 222–195 | 38°47′31.2″N 1°52′43.7″E | 38°47′28.7″N 1°52′31.8″E | AM | Carbonated rocks with encrusting sponges and gravels |
| 28 | 33 | 2020 | RD | 135–140 | 38°45′56.5″N 1°51′51.5″E | 38°46′3.7″N 1°51′45.7″E | AM | Rhodolith bed and rocks with sponges |
| 43 | 34 | 2020 | RD | 118–116 | 38°44′25.1″N 2°30′40.3″E | 38°44′26.9″N 2°30′33.5″E | EB | Rhodolith bed and rocks with sponges |
| 46 | 35 | 2020 | RD | 280–306 | 38°42′21.6″N 2°30′44.3″E | 38°42′31.3″N 2°30′42.5″E | EB | Basaltic rocks and fossil Ostreids with encrusting sponges |
| 94 | 36 | 2020 | GOC-73 | 142 | 39°1′13.8″N 2°51′2.5″E | 39°2′16.8″N 2°49′43.7″E | SW Ca | Fishing ground, sedimentary bottom |
| 07_1 | 37 | 2020 | ROV | 249–122 | 38°45′44.7″N 1°46′0.8″E | 38°45′22.3″N 1°46′22.1″E | AM | Sedimentary slope and rhodolith bed with sponges |
| 13 | 38 | 2020 | ROV | 465–352 | 38°48′22.3″N 1°52′57″E | 38°48′26.3″N 1°52′39.4″E | AM | Rocky slope with large sponges |
| 23 | 39 | 2020 | ROV | 133–169 | 38°44′27.6″N 2°29′15″E | 38°44′40.2″N 2°29′43.4″E | EB | Rocky slope, rhodolith bed with sponges and corals |
| 24 | 40 | 2020 | ROV | 150–134 | 38°44′46″N 2°29′28.3″E | 38°44′57.5″N 2°29′54.2″E | EB | Rocky slope and summit, rhodolith bed with sponges and corals |

**Note:**
($R_{survey}$) reference number in the survey. ($R_{study}$) correspondent reference in the present study. (GOC-73) experimental bottom trawl net. (BT) beam trawl. (DR) rock dredge. (ROV) Remote Operated Vehicle Liropus 2000. (SO) Ses Olives. (AM) Ausias March. (EB) Emile Baudot. (E Me) eastern Menorca. (SW Ca) south-western Cabrera Archipelago.

of EB there is the so-called Emile Baudot scarpment, which descend down to 2,600 m deep and connects the EB to the abyssal plain of the Algerian sub-basin (between the Balearic Isands and the Algerian coast) (*Acosta et al., 2004*).

The Algerian sub-basin hydrodynamics are mainly affected by density gradients, receiving warm and less saline Atlantic waters (*Pinot, López-Jurado & Riera, 2002*). These

surface waters have high seasonal temperature variation, ranging from 13 °C during winter to 26 °C during summer, when a strong vertical temperature gradient is established between 50 and 100 m deep. The water column below this depth shows fewer variations than in other parts of the western Mediterranean Sea, being mainly influenced by the Levantine Intermediate Water (LIW). This water mass, originated in the eastern Mediterranean, has temperature and salinity around 13.3 °C and 38.5 ppt, respectively, and is situated approximately between 200 and 700 m deep, just above the Western Mediterranean Deep Water, which is located in the lowest part of the water column (*Monserrat, López-Jurado & Marcos, 2008*). The western Mediterranean Intermediate Water, characterized by lower temperature (~12.5 °C) because it is formed during winter in the Gulf of Lions by deep convection when sea-air heat flux losses are high enough, is found at 100–300 m deep, but does not reach the Mallorca Channel every year (*Monserrat, López-Jurado & Marcos, 2008*).

Within the general oligotrophy of the Mediterranean, the southern Balearic Islands waters in the Algerian sub-basin show more pronounced oligotrophy than waters of the Balearic sub-basin located north of the Archipelago, and above all than the adjacent waters off the Iberian Peninsula and the Gulf of Lions (*Estrada, 1996*; *Bosc, Bricaud & Antoine, 2004*). The lack of supply of nutrients from land runoff and the lower influence of shelf/slope fronts flowing along the Iberian Peninsula and the northern insular shelf edge could explain these differences (*Massutí et al., 2014*; and references cited therein).

## Sampling

Sponge samples were collected at SO, AM and EB seamounts with a Jennings type beam trawl (BT) of 2 and 0.5 m horizontal and vertical openings, respectively, and a 5 mm mesh size cod-end, a rock dredge (RD) and the Remote Operated Vehicle (ROV) Liropus 2000 with an extensible arm. Sampling was performed during INTEMARES research surveys carried out in 2018, 2019 and 2020 on board of the R/Vs *Angeles Alvariño* and *Sarmiento de Gamboa* (Fig. 1). Additional samples from trawl fishing grounds of the continental shelf off Mallorca and Menorca were collected during the MEDITS research surveys carried out in 2017, 2019 and 2020 using the bottom trawl net GOC-73 of 2.5–3 and 18–22 m vertical and horizontal openings, respectively and a 10 mm mesh size cod-end, on board the R/V *Miquel Oliver* (Fig. 1). The sampling strategy of the MEDITS surveys is detailed in *Bertrand et al. (2002)* and *Spedicato et al. (2019)*. BT and GOC-73 have been shown efective for sampling macro-benthic species of the epibenthic and nektobenthic communities of sedimentary bottoms, respectively (*Reiss, Kröncke & Ehrich, 2006*; *Fiorentini et al., 1999*; *Ordines & Massutí, 2009*). The SCANMAR and MARPORT systems were used to control the deployment and retrieval of both gears to the bottom. By contrast, RD and ROV were used for sampling rocky bottoms and steep slopes. A summary of sampling stations used in the present work can be found in Table 1.

On board, specimens were photographed and stored in absolute EtOH. External morphology, color and texture were annotated prior to conservation. Spicule preparations and histological sections were made according to the standard methods described by *Hooper (2003)*. All the specimens were deposited in the Marine Fauna Collection (http://www.ma.ieo.es/cfm/) based at the Centro Oceanográfico de Málaga (Instituto

Español de Oceanografía), with the numbers from CFM7356 to CFM7417 and CFM7450-CFM7451 (Table S1).

The electronic version of this article in Portable Document Format (PDF) will represent a published work according to the International Commission on Zoological Nomenclature (ICZN), and hence the new names contained in the electronic version are effectively published under that Code from the electronic edition alone. This published work and the nomenclatural acts it contains have been registered in ZooBank, the online registration system for the ICZN. The ZooBank LSIDs (Life Science Identifiers) can be resolved and the associated information viewed through any standard web browser by appending the LSID to the prefix http://zoobank.org/. The LSID for this publication is: (urn:lsid: zoobank.org:pub:47EC2384-A88C-4654-8425-A7A46BC47AC5). The online version of this work is archived and available from the following digital repositories: PeerJ, PubMed Central and CLOCKSS.

## Morphological descriptions

Spicules were observed with a Nikon S-Ke optical microscope and photographed with a CMOS digital camera. Images were processed using the Fiji software (*Schindelin et al., 2012*). Whenever possible, at least 30 spicules per spicule type were measured. Spicules measures are written as length: min-average-max × thickness: min-average-max μm. Tangential and transversal thick sections were made with a scalpel and, if necessary, dehydrated with alcohol, mounted in DPX and observed under a compound microscope. Aliquots of suspended spicules were transferred onto foil, air dried, sputter coated with gold and observed under a HITACHI S-3400N scanning electron microscope (SEM).

## Molecular analysis

DNA was extracted from a piece of choanosomal tissue (~2 cm$^3$) using the DNeasy Blood and Tissue Extraction kit (QIAGEN). Polymerase chain reaction (PCR) was used to amplify the Cytochrome C Oxidase subunit I (*COI*; DNA barcoding) and the C1-D2 domains of the *28S* ribosomal gen, with the universal primers LCO1490/HCO2198 (*Folmer et al., 1994*) and C1' ASTR/D2 (*Vân Le, Lecointre & Perasso, 1993*; *Chombard, Boury-Esnault & Tillier, 1998*), respectively. PCR was performed in 50 μl volume reaction (34.4 μl ddH20, 5 μl Mangobuffer, 2 μl DNTPs, 3.5 MgCl$_2$, 1 μl of each primer, 1 μl BSA, 0.1 μl TAQ and 2 μl DNA). The PCR thermal profile used for *COI* amplification was (94 °C/5 min; 37 cycles (94 °C/15 s, 46 °C/15 s, 72 °C/15 s); 72 °C/7 min). *28S* amplification was carried out as detailed in *Chombard, Boury-Esnault & Tillier (1998)*. PCR products were visualized with 1% agarose gel, purified using the QIAquickR PCR Purification Kit (QIAGEN) and sequenced at Macrogen Inc. (South Korea).

Sequences were imported into BioEdit 7.0.5.2. (*Hall, 1999*) and checked for quality and accuracy with nucleotide base assignment. Sequences were aligned using Mafft (*Katoh et al., 2002*). The resulting sequences were deposited in the GenBank database (http://www.ncbi.nlm.nih.gov/genbank/) under the following accession numbers: MW858346–MW858351 and MZ570433 for *COI* sequences and MW881149–MW881153 for *28S* sequences; Table S1.

To assess the phylogeny of *Foraminospongia balearica* sp nov. and *Foraminospongia minuta* **sp. nov.**, two different approaches were used: Bayesian Inference (BI) and Maximum likelihood (ML). Here, we selected closely related sequences belonging to the orders Agelasida, Axinellida, Scopalinida and Biemnida, obtained after a BLAST search (*Altschul et al., 1990*). Additionally, two sequences belonging to the order Suberitida were used as outgroup. A complete list of the used sequences is available at Table S1. BI and ML analyses were performed with the CIPRES science gateway platform (http://www.phylo.org; *Miller, Pfeiffer & Schwartz, 2010*) using Mr Bayes version 3.6.2 (*Ronquist et al., 2012*) and RAxML (*Stamatakis, 2014*). For Mr Bayes, we conducted four independent Markov chain Monte Carlo runs of four chains each, with 5 million generations, sampling every 1000^th tree and discarding the first 25% as burn-in, while RAXML was performed under the GTRCAT model with 1000 bootstrap iterations. Convergence was assessed by effective sample size (ESS) calculation and was visualised using TRACER version 1.5. Genetic distance (p-distance) and number of base differences between pair of DNA sequences were estimated with MEGA version 10.0.5 software (*Kumar et al., 2018*).

# RESULTS

A total of 60 specimens belonging to two classes, nine orders, 13 families, 15 genera and 19 species were analyzed. All these species were collected at the Mallorca Channel seamounts, while three of them (*Phakellia robusta* Bowerbank, 1866, *Petrosia* (*Petrosia*) *raphida* Boury-Esnault, Pansini & Uriz, 1994 and *Hemiasterella elongata* (Topsent, 1928)) were also found at the continental shelf around Mallorca and Menorca. *In situ* images of some of these sponges, obtained with ROV from the seamounts of the Mallorca Channel, are shown in Fig. 2.

### Systematics
Phylum PORIFERA Grant, 1836
Class DEMOSPONGIAE Sollas, 1885
Suborder HETEROSCLEROMORPHA *Cárdenas, Perez & Boury-Esnault, 2012*
Order AGELASIDA Hartman, 1980
Family HYMERHABDIIDAE Morrow, Picton, Erpenbeck, Boury-Esnault, Maggs & Allcock, 2012

**Genus *Foraminospongia* gen. nov.**
(Figs. 2B, 2F, 2H, 3, 4, 5, 6; Table 2)

### Type species
*Foraminospongia balearica* **sp. nov.**

### Diagnosis
Hymerhabdiidae with massive, massive-tubular or bushy growth form, with styles, subtylostyles, tylostyles, and rhabdostyles. Besides, curved or angulated oxeas may be present. Ectosome with an aspicular dermal membrane supported by a plumoreticulated skeleton of styles, subtylostyles and tylostyles. Pores grouped into inhalant areas. Choanosome confusedly plumoreticulated.

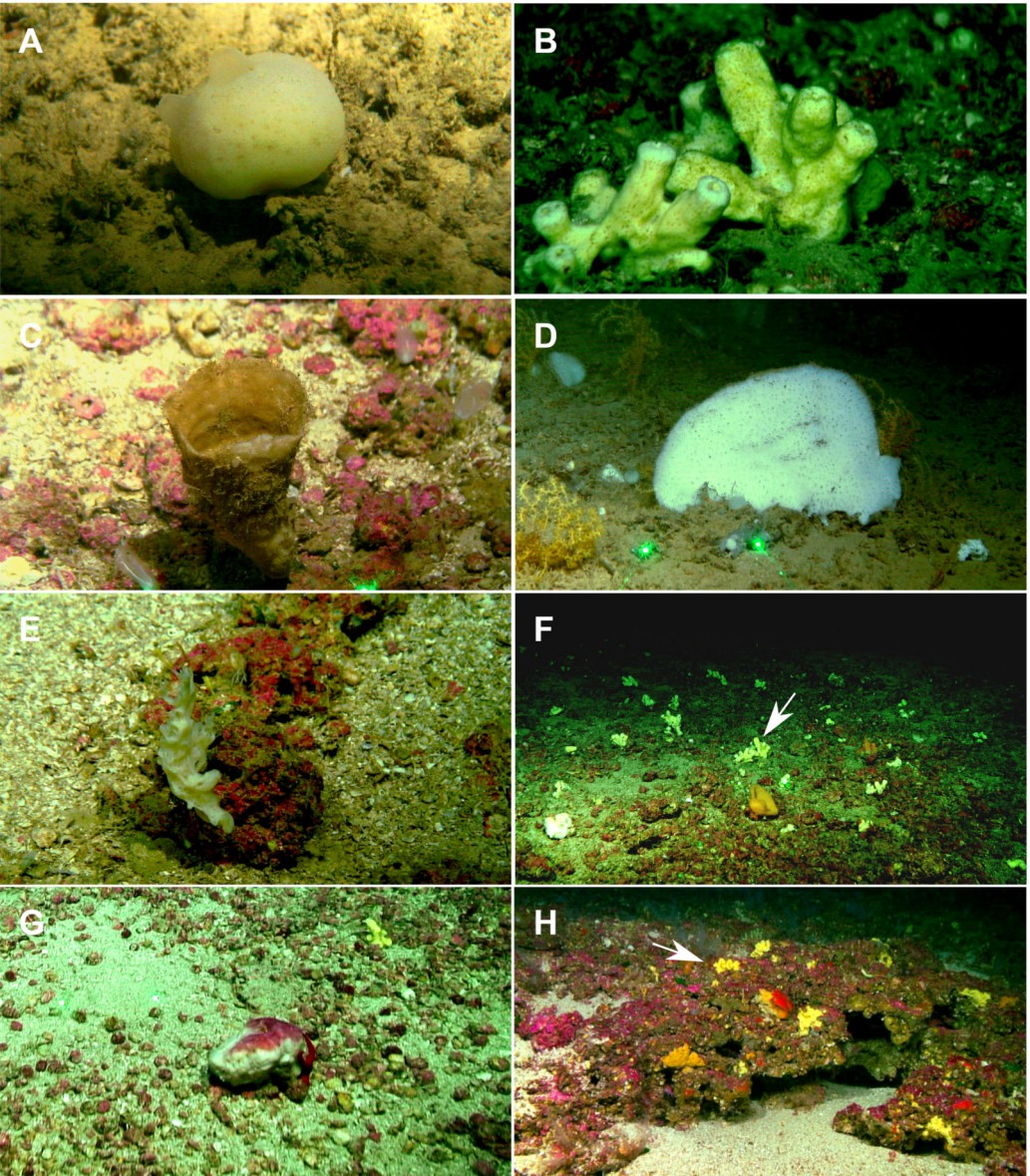

**Figure 2 Remote Operated Vehicle (ROV) images of the sponge fauna from the seamounts of the Mallorca Channel, Ses Olives (SO), Ausias March (AM) and Emile Baudot (EB).** (A) Specimen of *Polymastia polytylota* collected at 409 m depth in AM. (B) Holotype of *Foraminospongia balearica* **sp. nov.** collected at 129 m depth in the AM summit. (C) Specimen of *Phakellia ventilabrum* collected at 132 m depth in the EB summit. (D) Uncollected specimen of *Phakellia* sp. at 374 m depth in the north knoll of AM. (E) Specimen of *Haliclona (soestella) fimbriata* collected at 131 m depth in the EB. (F) Rhodolith bed at 110 m depth in the summit of AM, with different sponge species, including *F. balearica* **sp. nov.** (arrow), (G) uncollected specimen of *Calyx* cf. *tufa* at 106 m depth in the summit of AM, (H) coralligenous bottom at 97 m depth in the summit of AM, with several sponges, including *F. balearica* **sp. nov.** (arrow).

## Etymology

From the Latin *foramen* (pores) and *spongia* (sponge). The name refers to the fact that in both species, their skin has areas where pores are grouped, giving a characteristic macroscopical appearance.

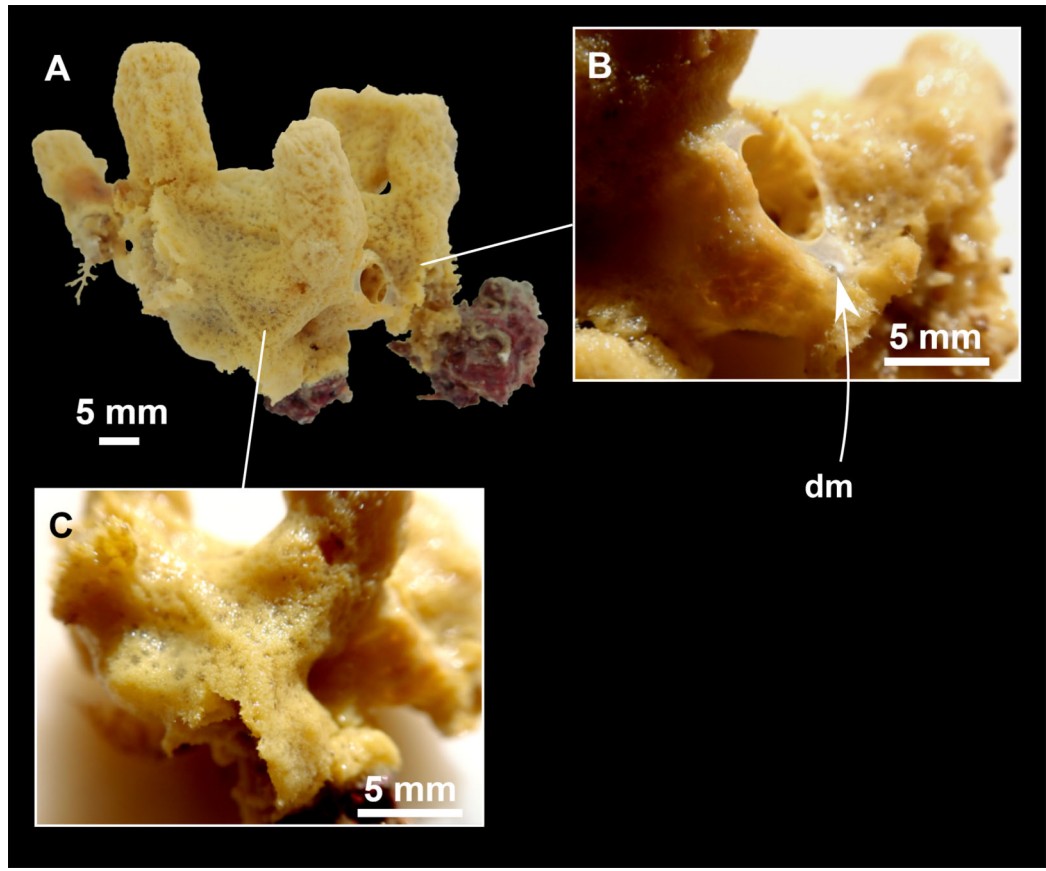

**Figure 3** *Foraminospongia balearica* **sp. nov.** (A) Habitus of CFM-IEOMA-7356/i802 (holotype) in fresh state, with (B) Detail of the oscula and the dermal membrane (dm) and (C) Macroscopic view of the grooves at the skin.

*Foraminospongia balearica* **sp. nov.**

(Figs. 2B, 2F, 2H, F3, 4, 5; Table 2)

**Diagnosis**

Massive-tubular to bushy *Foraminospongia*, with styles, rhabdostyles and oxeas.

**Etymology**

The name refers to the Balearic Islands, the area where the species has been collected.

**Material examined**

*Holotype*: CFM-IEOMA-7356/i802, St 37, AM, ROV.

*Paratypes*: CFM-IEOMA-7357/i144, St 4, EB, BT; CFM-IEOMA-7358/i293_1, St 9, AM, BT; CFM-IEOMA-7359/i239 (not described), St 8, AM, BT; CFM-IEOMA-7360/i745 (not described), St 26, EB, BT; CFM-IEOMA-7361/i824_4, St 39, EB, ROV.

*Specimens observed but not sampled:* St 12, EB, BT; St 14, EB, BT.

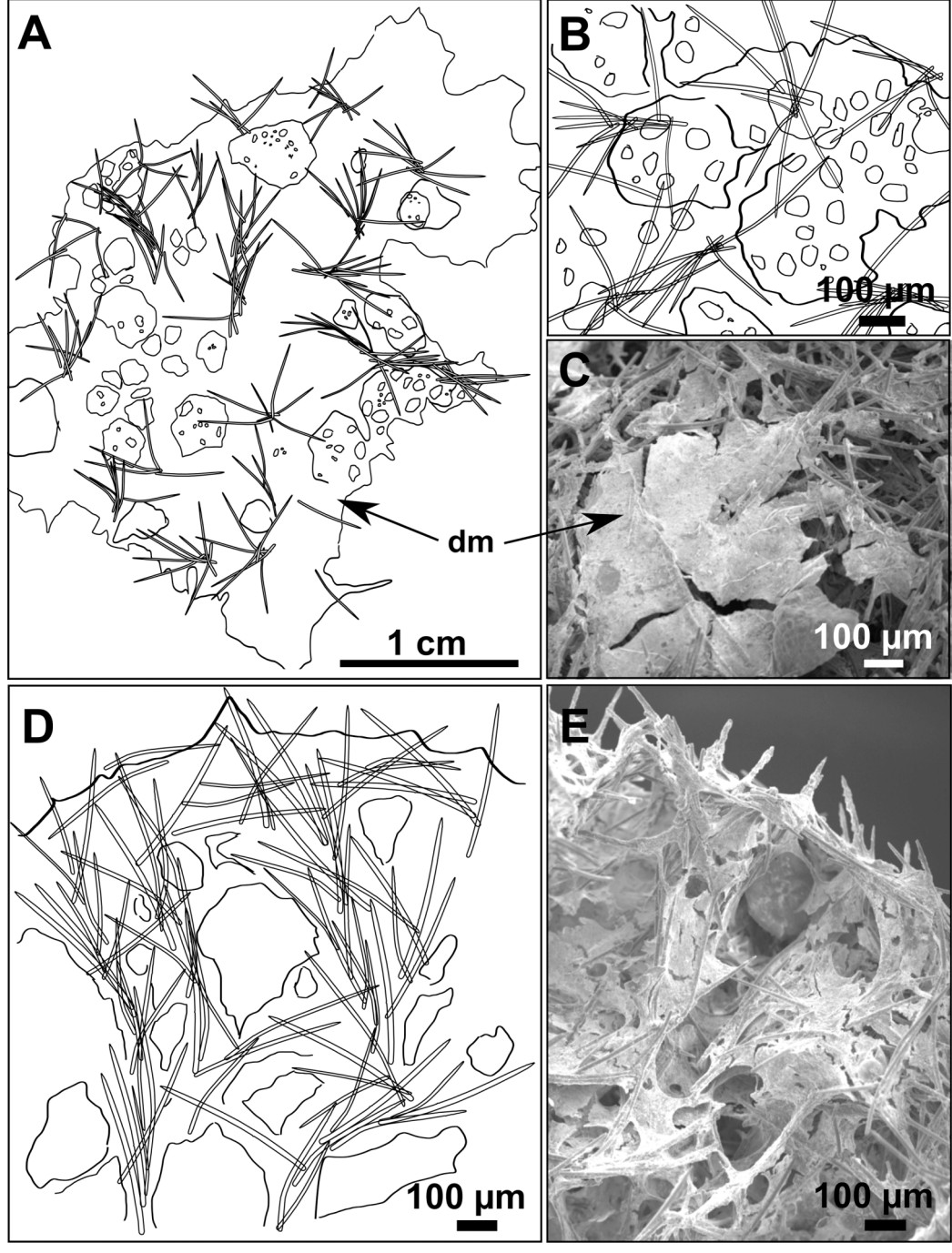

**Figure 4 Skeletal arrangement of *Foraminospongia balearica* sp. nov., CFM-IEOMA-7356/i802 (holotype).** (A–C) Tangential images of the surface, showing the dermal membrane (dm). (D–E) Transversal sections.

## Comparative material

*Foraminospongia minuta* **sp. nov.**: CFM-IEOMA-7362/i439, St 27, RD, SO; CFM-IEOMA-7363/i474, St 29, SO, RD.

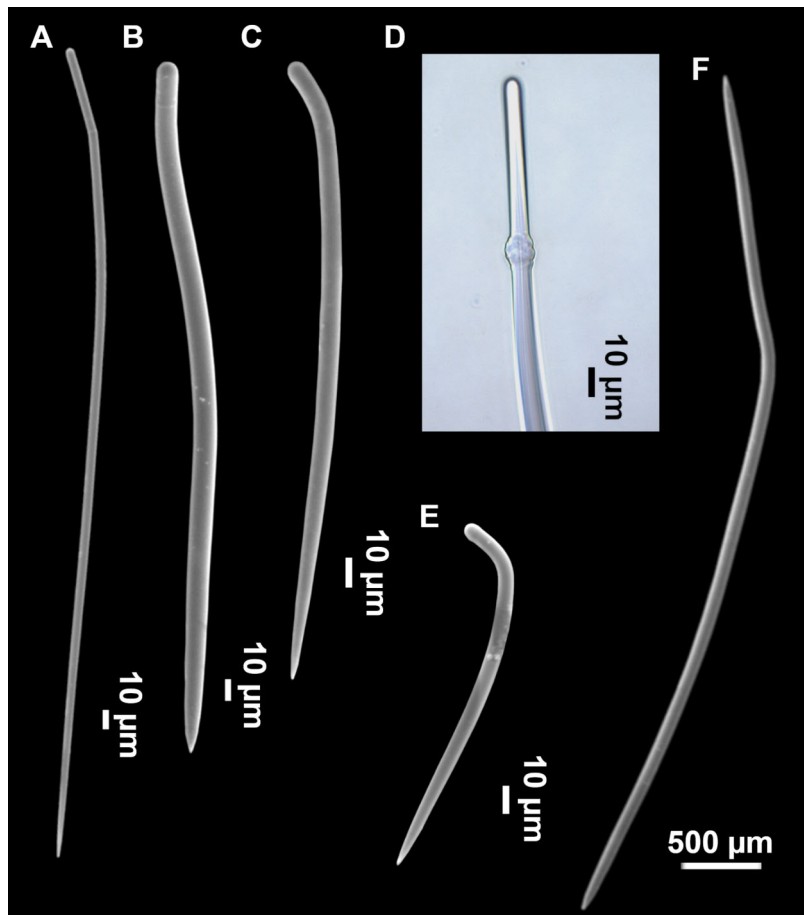

**Figure 5 SEM images of the spicules from *Foraminospongia balearica* sp. nov. CFM-IEOMA-7358/ i293_1 (paratype).** (A–D) Styles. (E) Rhabdostyles. (F) Oxea.

*Rhabderemia* sp.: CFM-IEOMA-7415/i729_1 (only a slide deposited at the CFM-IEOMA), St 35, EB, RD.

### Description

Massive-tubular or bushy sponges (Figs. 2B, 2F, 2H and 3A). Largest specimens up to 6 cm in diameter. When present, tubes are 2–3 cm in height and 1 cm in diameter. Sometimes several tubes are fused on another of its sides. Consistency slightly elastic, brittle, easily broken when manipulated. Surface smooth, rough to the touch. Color in life golden yellow, tan after preservation in EtOH. A translucid membrane is present, more evident near the oscula (Fig. 3B). Subdermal grooves forming a visible pattern (Fig. 3C). Circular oscula 0.3–0.6 cm. In most cases, oscula are placed at the end of tubes, however, the holotype also has a large osculum in the main body (Fig. 3B).

### Skeleton

Ectosome characterized by a plumoreticulated tangential skeleton and a dermal membrane (Figs. 4A, 4B and 4C). In some areas of the dermal membrane there are small

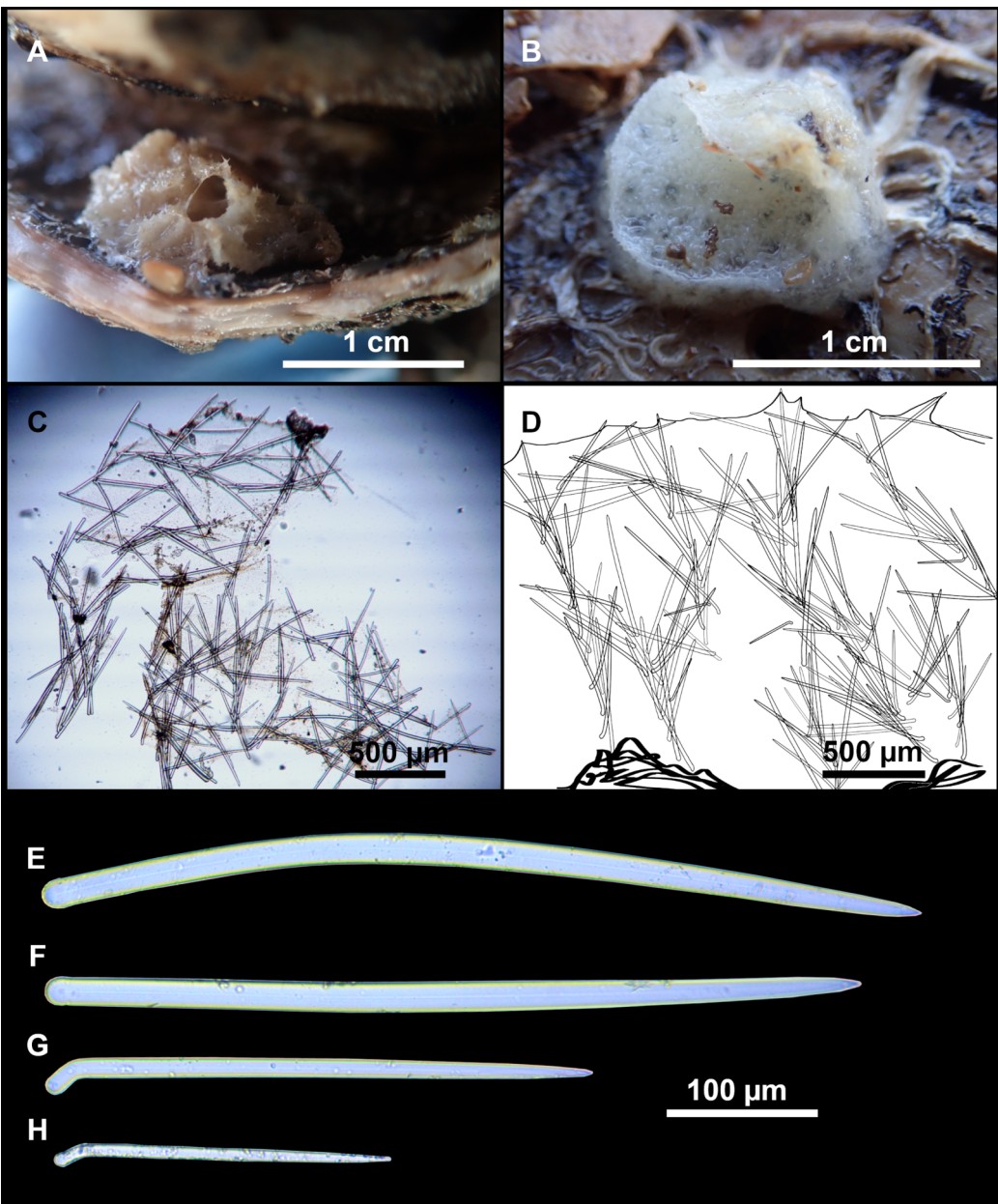

**Figure 6** *Foraminospongia minuta* **sp. nov.** (A) Habitus of CFM-IEOMA-7362/i439 (holotype) on fresh state. (B) On deck image of CFM-IEOMA-7363/i474 (paratype). (C) Optic microscope image of the tangential skeleton of the holotype. (D) Schematic illustration of the choanosome of the holotype. (E–F) Styles. (G–H) Rhabdostyles.

pores gathered. These porae areas correspond to the grooves that are perceptible to the eye. Choanosome, confusedly plumoreticulated with extensive spaces and ascending spicule tracts of 2–5 styles, sometimes protruding the surface. The tracts contain abundant spongin. In between the tracts transversal spicules are abundant (Figs. 4D–4E).

**Table 2 Comparative characters of *Foraminospongia balearica* sp. nov. and *Foraminospongia minuta* sp. nov.**

| Specimen | Style | Rhabdostyle | Oxea | Depth | Area |
|---|---|---|---|---|---|
| *Foraminospongia balearica* **sp. nov.** | | | | | |
| CFM-IEOMA-7356/i802 Holotype | 188–378–492 × 6–11–14 | 90–179 × 4–7 (*n* = 9) | 456–609 × 9–11 (*n* = 3) | 249–122 | AM St 13 |
| CFM-IEOMA-7357/i144 Paratype | 197–378–501 × 4–9–12 | 108–164 × 3–5 (*n* = 5) | 249–493–656 × 4–8–12 (*n* = 15) | 128 | EB St 4 |
| CFM-IEOMA-7358/i293_1 Paratype | 179–356–516 × 3–8–14 | 138–179 × 3–6 (*n* = 5) | 328–527–763 × 3–8–13 | 127 | AM St 9 |
| CFM-IEOMA-7361/i824_4 Paratype | 177–403–634 × 5–9–13 | 92–165 × 3–6 (*n* = 9) | 600 × 9 (*n* = 1) | 133–169 | EB St 39 |
| *Foraminospongia minuta* **sp. nov.** | | | | | |
| CFM-IEOMA-7362/i439 Holotype | 283–509–658 × 9–14–21 | 175–262 × 7–9 (*n* = 7) | np | 318–288 | SO St 26 |
| CFM-IEOMA-7363/i474 Paratype | 244–416–555 × 10–14–20 | 147–232 × 7–9 (*n* = 4) | np | 315–295 | SO St 28 |

**Note:**
Depth (m), area (SO, Ses Olives; AM, Ausias March; EB, Emile Baudot) and sampling station (St; see R*study* in Table 1) where these specimens were collected are also shown. Spicule measures are given as minimum-mean-maximum for total length × minimum-mean-maximum for total width. A minimum of 30 spicules per spicule kind are measured, otherwise it is stated. All measurements are expressed in μm. Specimen codes are the reference numbers of the CFM-IEOMA/and author collection. np, not present.

**Spicules**

Styles (Figs. 3A–3D): Fusiform, most gently curved, but sometimes abruptly curved once or twice. When the curvature is in the last portion of the spicule, they may resemble rhabdostyles. Roundish heads and sharp tips, sometimes telescoped, strongylote forms present. Swellings may happen at the head or below, sometimes barely visible, sometimes more patent, rarely tuberculated (Fig. 3D). Size range constant between specimensspecimens, not influenced by depth nor area (Table 2). They measure 177–375–634 × 3–9–14 μm.
Rhabdostyles (Fig. 3E): Uncommon. Abruptly curved below the head. Stylote, subtylote and tylote modifications present. Round head and acerated tips. They measure 90–143–179 × 3–5–7 μm. specimens
Oxeas (Fig. 3F): specimensCurved or bent, with one, two or several curvatures, sometimes slightly sinuous. Tips acerated or telescoped. They measure 249–520–763 × 3–8–13 μm. Their abundance varies between specimens.

**Genetics**

Two *COI* Folmer fragment sequences were obtained for the Holotype (CFM-IEOMA-7356/i802) and for one paratype (CFM-IEOMA-7358/i293_1) (Genbank id's MW858346 and MW858347, respectively). Besides, we obtained a *28S* sequence (C1-D2 domains) for the Holotype (GenBank id MW881153).

**Ecological notes**

The species is very abundant on the EB and AM, between 100 and 169 m (Table 2). It can be mainly found on rhodolith beds and sedimentary bottoms with gravel, together with other sponges like *Poecillastra compressa* (Bowerbank, 1866), *Axinella* spp.,

*Halichondria* spp. or some Haplosclerids, as well as with a very broad number of crustaceans and echinoderms. It was also collected down to 433 m (St 14).

### *Foraminospongia minuta* sp. nov.
(Fig. 6; Table 2)

### Diagnosis
Small, massive-encrusting and greyish in color *Foraminospongia*, with only styles and rhabdostyles as spicules.

### Etymology
The name refers to the small size of the two collected specimens.

### Material examined
*Holotype*: CFM-IEOMA-7362/i439, St 27, SO, RD.
*Paratype*: CFM-IEOMA-7363/i474, St 29, SO, RD.

### Comparative material
*Foraminospongia balearica* **sp. nov.**: CFM-IEOMA-7357/i144, St 4, EB, BT; CFM-IEOMA-7358/i293_1, St 9, AM, BT; CFM-IEOMA-7356/i802, St 37, AM, ROV; CFM-IEOMA-7361/i824_4, St 39, EB, ROV.
*Rhabderemia* sp.: CFM-IEOMA-7415/ i729_1, St 35, EB, RD.

### Description
Small massive-encrusting sponge (Figs. 6A and 6B), about 1.5 cm in diameter and 0.5 cm in height. Consistency: compressible and slightly crumby. Velvety surface. The holotype was brownish due to mud, the paratype was greyish, both in life and after preservation in EtOH. Translucent membrane that can be peeled off is present, with grooves forming a distinguishable pattern (Fig. 6B). A single, circular oscule is present on the holotype.

### Skeleton
The ectosome consists of a tangential reticulation of styles (Fig. 6C), and some loose rhabdostyles.

The choanosome is a plumoreticulated net of styles, with some loose rhabdostyles (Fig. 6D).

### Spicules
Styles (Figs. 6E and 6F): Fusiform, gently curved or straight. Heads roundish and swelled in most cases. Sharp tips. Most are tylota. Size range variable between the holotype and the paratype (Table 2). They measure 244–465–658 × 9–14–21 μm.

Rhabdostyles (Figs. 6G and 6H): Uncommon. Abruptly curved below the head, most with roundish, tylota modifications at the head and sharp tips. They measure 147–209–262 × 7–8–9 μm.

### Genetics

Sequences of *COI* Folmer fragment and *28S* C1-D2 domains were obtained for the holotype and deposited in Genbank under accession numbers MW858348 and MW881151, respectively.

### Ecological notes

Both specimens were found at SO, between 288 and 318 m deep, associated to hard bottoms with fossil ostreids reefs.

### Remarks on *F. balearica* sp. nov. and *F. minuta* sp. nov.

Regarding the interspecific variability of *F. balearica* **sp. nov.**, the spicules of the studied specimens are in the same size range, except for the styles of the specimen from AM (CFM-IEOMA-7358/i293_1), which are shorter and thinner than those of the specimens from EB. Also, specimen CFM-IEOMA-7358/i293_1 has much more abundant oxeas than the others.

Regarding *F. minuta* **sp. nov.**, the features of this species support the differential diagnostic characters of the genus *Foraminospongia* (plumoreticulated choanosomal skeleton, ectosome formed by a reticulation of spicules, dermal aspicular membrane with poral areas, presence of large styles and small rhabdostyles), but differs from *F. balearica* **sp. nov.** in its external morphology, being much smaller and massive-encrusting compared to massive-tubular or bushy and of a greyish color instead of golden yellow in the latter. Also, the spicular complement is different: *F. minuta* **sp. nov.** lacks oxeas and has longer and thicker styles and rhabdostyles. The differences in the size of the styles between the holotype and the paratype are notable, considering that both were collected at similar depths and habitats. These differences could suggest intraspecific variability for the spicule size within the species; however, more specimens are needed to corroborate this statement.

The morphological differences between the two species are backed by genetic results. The phylogenetic reconstructions for *COI* and *28S* fragments show well-supported separation between the two *F. balearica* **sp. nov.** sequences and the *F. minuta* **sp. nov.** sequence. Between the two species, the differences in bp and p-distance (in percentage) for *COI* Folmer and the *28S* fragments were 1 bp/0.2% and 1bp/0.1%, respectively.

### Remarks on the genus *Foraminospongia*

The family Hymerhabdiidae was recently erected to include the genera *Hymerhabdia*, *Prosuberites* and some species of the polyphyletic genus *Axinella* and *Stylissa* (*Morrow et al., 2019*). Here, we propose *Foraminospongia* as a new hymerhabdiid genus. The main differences between *Foraminospongia* gen. nov. and both *Hymerhabdia* and *Prosuberites* are the growing habit, with *Foraminospongia* gen. nov. being massive, massive-tubular or bushy against encrusting. Also, it differs from *Prosuberites* in the presence of rhabdostyles and oxeas. However, the presence of rhabdostyles and oxeas is shared with *Hymerhabdia*, but the genetic differentiation between *Foraminospongia* and *H. typica* (type species of *Hymerhabdia*) is clear (Fig. 7). In addition, the ectosome with a dermal membrane and grouped pore areas of *Foraminospongia* is not present in any

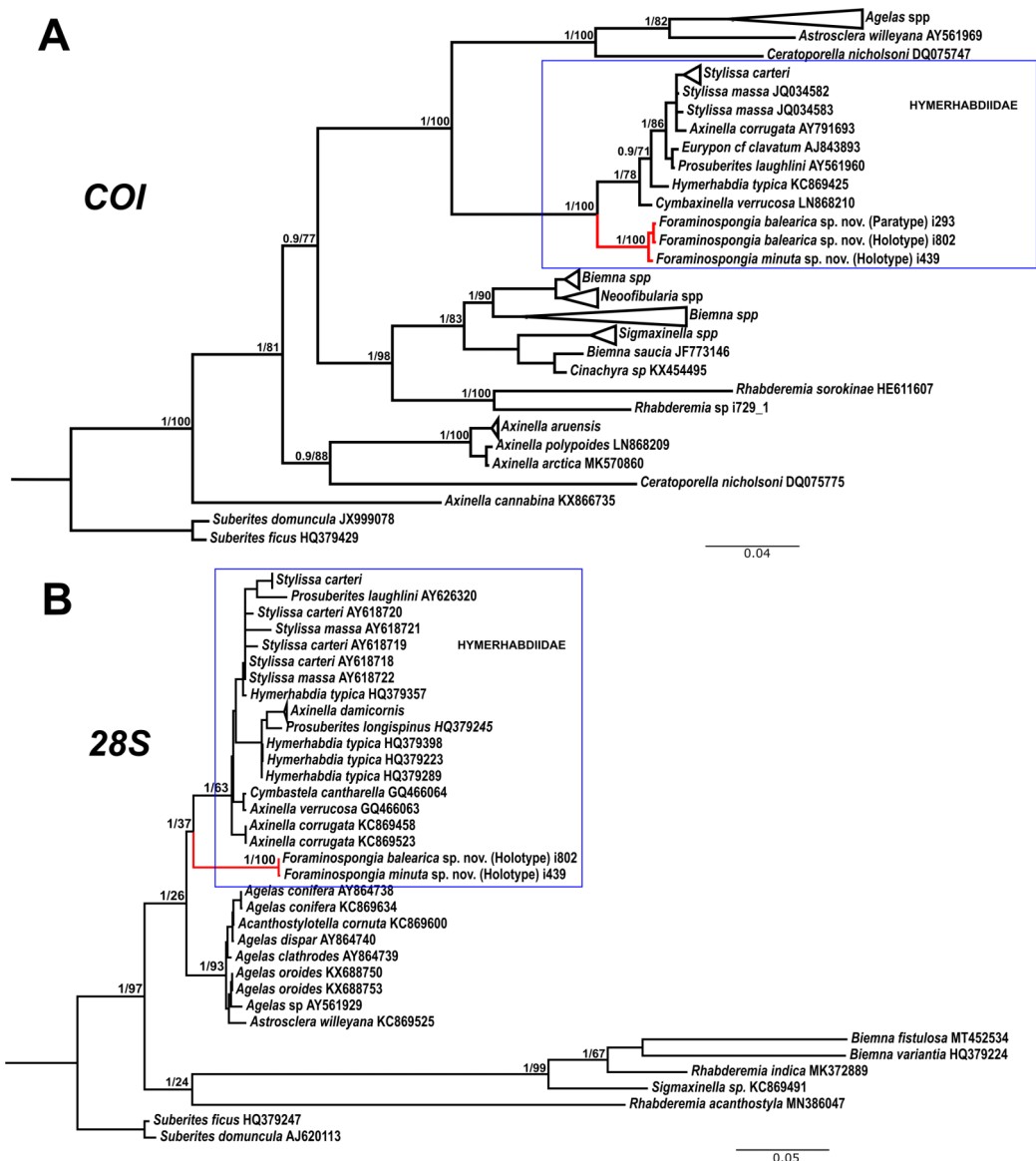

**Figure 7 Phylogenetic tree topology for specimens of *Foraminospongia balearica* sp. nov., *Foraminospongia minuta* sp. nov. described in the present study and other related Agelasids.** The three was constructed with Maximum likelihood and Bayesian inference, based on *COI* (A) and *28S* (B) fragments. Posterior probabilities and bootstrap support values are shown at the nodes. A sequence of *Suberites domuncula* and *Suberites ficus* are used as outgroups in both trees.

*Hymerhabdia* apart *Hymerhabdia oxeata* (*Dendy, 1924*) that has a dermal membrane, although neither Dendy nor the re-examination done by *Van Soest & Hooper (1993)* described pore areas. Therefore, *H. oxeata* could represent an intermediate stage between genuine *Hymerhabdia* and *Foraminospongia* species. However, the last statement is only speculative and must be checked in future works.

As stated before, there are species of *Axinella* and *Stylissa* that are grouped inside Hymerhabdiidae. Although currently all these species are kept in Axinellida and

Suberitida, respectively (*Van Soest et al., 2021*), they are phylogenetically related to *Foraminospongia* (see Fig. 7). To resolve this relatedness, we have included in the phylogenetic analysis sequences of *A. damicornis* (*Esper, 1794*), *A. verrucosa* (*Esper, 1794*), *A. corrugata* (*George & Wilson, 1919*), *S. carteri* (*Dendy, 1889*) and *S. massa* (*Carter, 1887*) used by *Morrow et al. (2012)* to define Hymerhabdiidae. The resulting trees show that those species are clearly different from *Foraminospongia*, which is corroborated by their morphology (*Pansini, 1984*; *Hooper & Van Soest, 2002*).

The genus *Rhabderemia* (Order Biemnida, family Rhabderemiidae) resembles *Foraminospongia* in having rhabdostyles and possessing a plumoreticulated choanosomal skeleton. However, most Rhabderemia also have peculiar rugose microscleres (thraustoxeas, spirosigmata, thraustosigmata, microstyles). To clarify the potential relatedness of *Rhabderemia* and *Foraminospongia*, we have included in the phylogenetic analyses the species *Rhabderemia sorokinae* Hooper, 1990, *R. indica* Dendy, 1905 and *R. destituta Van Soest & Hooper, 1993*. Moreover, we included in the *COI* tree one sequence of an encrusting *Rhabderemia* sp. (CFM-IEOMA-7415/ i729_1; Genbank ID MZ570433) collected at the EB, with spined rhabdostyles, toxas and spirosigmata (Fig. 7B). Other sequences of Biemnida available at the genbank have also been included (see Table S1).

The sequence of *Rhabderemia* sp. (CFM-IEOMA-7415/ i729_1; Genbank ID MZ570433) clustered together with *R. sorokinae*, a Great Barrier Reef sponge which also has spined rhabdostyles, toxas and spirosigmata, in addition to microspined microstyles, a fact that confirms that archetypical rhabderemids are not related to *Foraminospongia*. However, microscleres are lacking in *R. mona* (*de Laubenfels, 1934*) and *R. destituta*, so they resemble *Foraminospongia*. *Rhabderemia mona* is a Caribbean sponge described from bathyal depths off Puerto Rico, used to erect the genus *Stylospira* for "sponges having no spicules other than peculiar spirally twisted styles" (*de Laubenfels, 1934*). This single specimen was later studied by *Van Soest & Hooper (1993)* on a revision of the genus, who concluded that *Stylospira* should be considered a subgenus of *Rhabderemia*. *Van Soest & Hooper (1993)* also described *R. destituta* from the Galapagos Islands, a second species matching de Laubenfels' diagnosis. Interestingly, apart from the lack of any kind of microscleres (even though de Laubenfels reported raphides for *R. mona*, not found by *Van Soest & Hooper, 1993*), both species had smooth rhabdostyles, just as *Foraminospongia*, which is in contrast to most of the other *Rhabderemia* spp. Among the 30 known species of the genus, only *R. stellata* (*Bergquist, 1961*), *R. spirophora* (*Burton, 1931*), *R. gallica* (*Van Soest & Hooper, 1993*), *R. profunda* (*Boury-Esnault, Pansini & Uriz, 1994*), *R. africana Van Soest & Hooper, 1993*, *R. prolifera* Annandale, 1915 and *R. meirimensis Cedro, Hajdu & Correia, 2013* have smooth rhabdostyles.

Unfortunately, there are no sequences available for *R. mona* nor *R. destituta*, so their potential relatedness with *Foraminospongia* cannot be addressed. However, it should be noted that both species have only rhabstosytles as megascleres, wich is in contrast to the heterogenous set of megascleres shown by *Foraminospongia* (styles, tylostyles, subtylostyles, rhabdostyles and oxeas). This seems a strong argument against congeneric relatedness with *Foraminospongia*. However, this issue should be properly addressed in the future when sequences of *R. mona* and *R. destituta* become available.

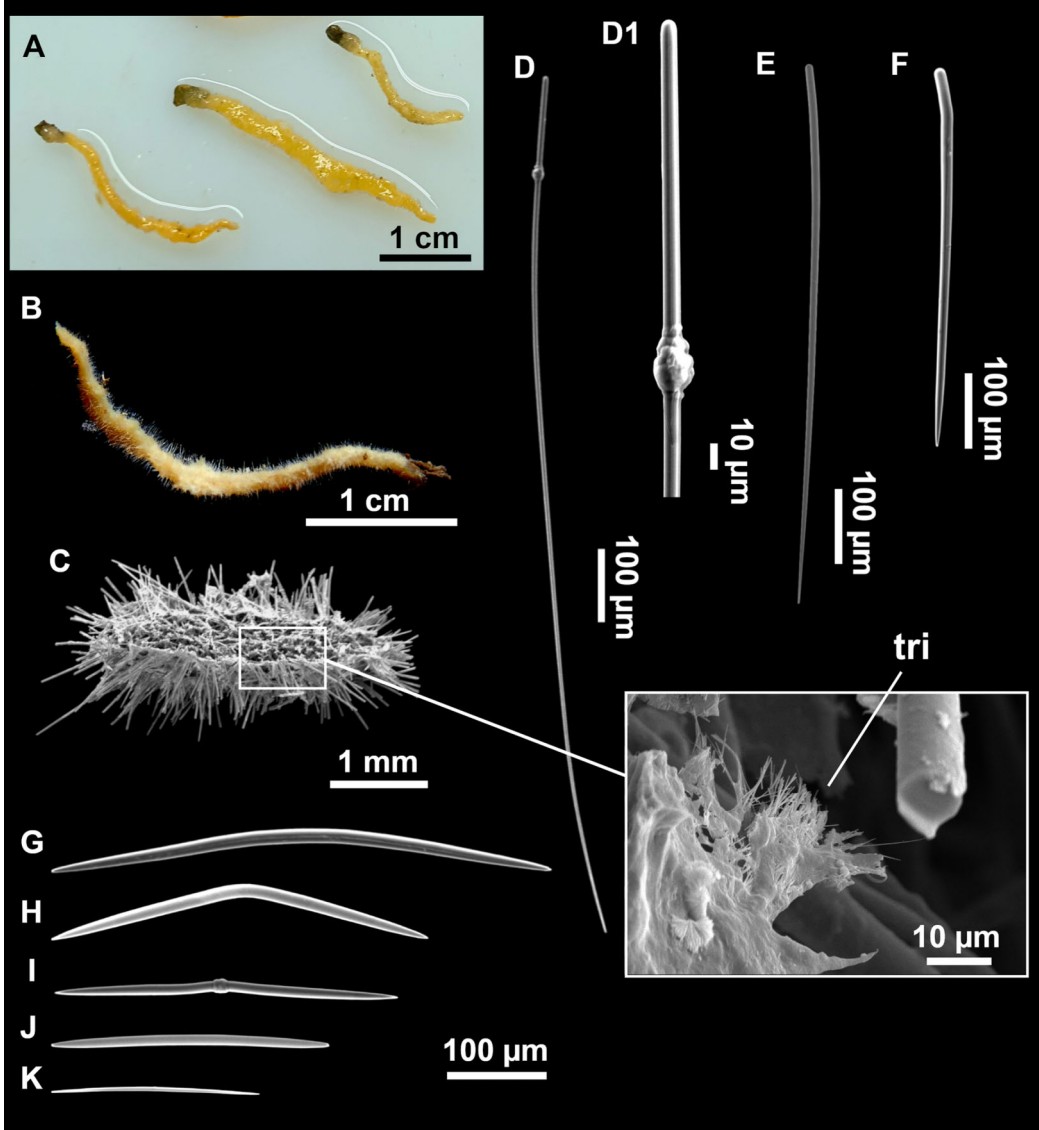
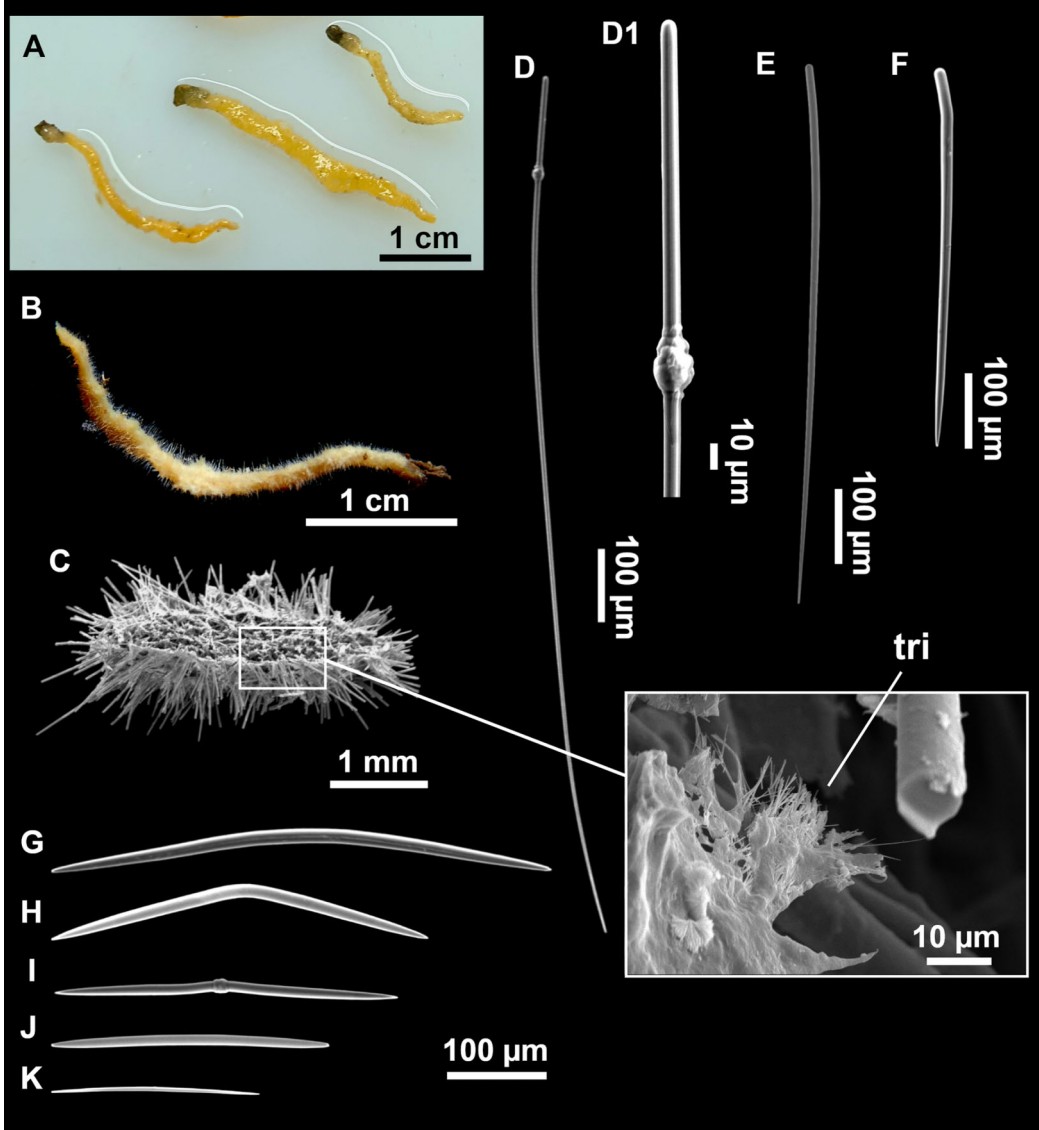

**Figure 8** *Axinella spatula Sitjà & Maldonado, 2014.* (A) Photograph of fresh material deposited under CFM-IEOMA-7364-7366/i338_1A–1C. (B) Habitus of CFM-IEOMA-7366/i338_1C preserved in EtOH. (C) SEM images of the skeletal structure of CFM-IEOMA-7366/i338_1C with detail of the inner ecto-somal layer, with trichodragmata (tri). (D) Long styles with (D1) subterminal swelling. (E) Regular shaped style. (F) Style with rhabdose modification. (G) Oxea asymmetrically curved. (H) Oxea centrocurved. (I) Oxea centrotylota. (J–K) Small oxeas.

Order AXINELLIDA Lévi, 1953
Family AXINELLIDAE Carter, 1875
Genus *Axinella* Schmidt, 1862
***Axinella spatula* Sitjà & Maldonado, 2014**
(Fig. 8; Table 3)

**Table 3 Comparative characters of the collected specimens of *Axinella spatula Sitjà & Maldonado, 2014*, and those reported for the type material (*Sitjà & Maldonado, 2014*).**

| Specimen | Styles | Oxeas | Trichodragmata | Color | Depth | Area |
|---|---|---|---|---|---|---|
| MNCN-Sp145-BV33A *Sitjà & Maldonado (2014)* Holotype | 165–1050 × 3–15 | 180–520 × 2.5–15 | 25–30 × 5–8 | Beige after EtOH | 134–173 | Alboran Island |
| MNCN-Sp188-BV41A *Sitjà & Maldonado (2014)* Paratype | 119–1400 × 4–15 | 190–750 × 5–20 | 25–35 × 5–8 | Beige after EtOH | 102–112 | Alboran Island |
| MNCN-Sp57-BV21B *Sitjà & Maldonado (2014)* Paratype | 245–1225 × 8–18 | 120–432 × 9–12 | 25–30 × 6–10 | Black after EtOH | 93–101 | Alboran Island |
| CFM-IEOMA-7364/i338_1A This work | 349–613–1161 × 7–13–16 (n = 20) | 187–374–507 × 5–11–16 | 32–39–47 × 5–7–10 | Orange in life orange beige after EtOH | 152 | EB St 11 |
| CFM-IEOMA-7365/i338_1B This work | 248–900–1304 × 11–17–26 (n = 17) | 219–377–485 × 7–11–16 | 36–45–56 × 5–7–8 (n = 9) | Orange in life orange beige after EtOH | 152 | EB St 11 |
| CFM-IEOMA-7366/i338_1C This work | 332–638–1265 × 4–12–17 (n = 23) | 247–332–493 × 7–10–16 | 32–39–52 × 5–7–11 | Orange in life orange beige after EtOH | 152 | EB St 11 |

**Note:**
Depth (m), area (EB, Emile Baudot) and sampling station (St; see R*study* in Table 1) where these specimens were collected are also shown. Spicule measures are given as minimum-mean-maximum for total length × minimum-mean-maximum for total width (or as they appear in the cited texts). A minimum of 30 spicules per spicule kind are measured, otherwise it is stated. All measurements are expressed in µm. Specimen codes are the reference numbers of the CFM-IEOMA/and author collection for the Balearic specimens and the reference numbers of Invertebrate Collection of the National Museum of Natural Sciences (MNCN) of Madrid for *Sitjà & Maldonado (2014)* specimens.

**Material examined**

CFM-IEOMA-7364/i338_1A, CFM-IEOMA-7365/i338_1B and CFM-IEOMA-7366/i338_1C, St 11, EB, BT.

**Description**

Small, erect, cylindrical, and slightly flattened sponges, up to 3 cm height and 2–3 mm width (Figs. 8A–8C). Very hispid all along the body. Orange in life (Fig. 8A) and orange beige after preservation in EtOH (Fig. 8B).

**Skeleton**

As in *Sitjà & Maldonado (2014)*.

**Spicules**

Megascleres

Styles (Figs. 8D–8F): with a wide size range, rounded ends and sharp tips. Straight or slightly curved. The largest ones may be slightly sinuous, sometimes with subterminal swellings (Fig. 8D1). Rhabdostyle modifications are present in small and intermediateintermediate stages (Fig. 8F). They measure 248–722–1304 × 4–14–17 µm.

Oxeas: curved or bent, sometimes centrotylote (Figs. 8G–8K), with the curvature point at the center or displaced towards one of the extremities. Tips acerated. They measure 187–357–507 × 5–11–16 µm.

Microscleres

Raphides in trichodragmata (Fig. 8C, detail), abundant and of the same morphology in all specimens. They measure 32–40–56 × 5–7–11 µm.

## Ecology notes

Found only on the north-eastern part of EB, at 152 m deep, on gravel bottoms with dead rhodoliths and with a large abundance of sponges such as *P.* (*Petrosia*) *ficiformis* (Poiret, 1789), *P.* (*Petrosia*) *raphida* *Boury-Esnault, Pansini & Uriz, 1994*, *P.* (*Strongylophora*) *vansoesti* *Boury-Esnault, Pansini & Uriz, 1994* and several Tetractinellida.

## Remarks

The specimens match well with those originaly described from the Alboran Sea. Balearic specimens are smaller (maximum height of 3 cm against maximum height of 10 cm in alboran specimens). Also, the size range of their styles and oxeas are not as wide as in Alboran specimens and trichodragmata of our specimens were always longer (Table 3).

*Sitjà & Maldonado (2014)* described two phenotypes, according to the color acquired after preservation in EtOH (black or beige). Also, they found skeletal variations linked to each group, corresponding to a higher or lower presence of short styles, the morphology of the trichodragmata or the skeletal arrangement. The specimens collected here correspond only to the beige phenotype.

With the present record, the species distribution widens towards the north-western Mediterranean Sea, since previously it was known only for the type's location, at the Alboran Island (*Sitjà & Maldonado, 2014*).

## *Phakellia robusta* Bowerbank, 1866

**Synonymised names**.

*Phacellia robusta* (*Bowerbank, 1866*) (misspelling of genus name)

**Material examined**

CFM-IEOMA-7367/i347_2, St 12, EB, BT; CFM-IEOMA-7368/i405 and CFM-IEOMA-7369/i409, St 15, EB, BT; CFM-IEOMA-7370/i414_2, St 16, EB, BT; CFM-IEOMA-7371/i417, St 17, EB, BT; CFM-IEOMA-7372/i712, St 25, EB, BT; CFM-IEOMA-7373/i731, St 35, EB, RD; CFM-IEOMA-7374/POR760, St 20, south-western Cabrera Archipelago, GOC-73; CFM-IEOMA-7375/POR762, St 21, south-western Cabrera Archipelago, GOC-73.

**Ecology notes**

The species was frequent at the studied area, being found in a broad depth range (150–750 m) on both rocky and sedimentary bottoms. In the trawl fishing grounds of the continental shelf around Mallorca and Menorca it was mostly found below 300 m deep, where most of the collected specimens were larger. In the seamounts of the Mallorca Channel, the species was common on gravel bottoms 150–170 m deep, where specimens tended to be very small (1.5–3 cm in height) and in rocky outcrops and vertical walls, where sizes were intermediate (4–12 cm in height) and large (20–35 cm in height).

**Remarks**

The species is reported for the first time in the Mallorca Channel, being its second record at the Balearic Islands, where it was previously recorded by *Santín et al. (2018)* from

the Menorca Channel. In the Mediterranean, it is also known from the Gulf of Lions (*Vacelet, 1969*), the Tyrrhenian Sea (*Topsent, 1925*), the Alboran Sea (*Maldonado, 1992*), the Strait of Sicily (*Calcinai et al., 2013*) and the Adriatic Sea (*D'Onghia et al., 2015*). Besides, the species has been reported from several localities of the North Atlantic including the Gulf of Cadiz (*Sitjà et al., 2019*), the Azores Islands (*Topsent, 1904*), the Cantabrian Sea (*Ferrer Hernández, 1914*) and the North Sea (*Bowerbank, 1866*).

### *Phakellia ventilabrum* (*Linnaeus, 1767*)
**Synonymised names**
*Halichondria ventilabrum* (*Linnaeus, 1767*)
*Phacellia ventilabrum* (misspelling of genus name)
*Phakellia ventilabra* (ruling of ICZN)
*Spongia strigose* Pallas, 1766 (genus transfer & junior synonym)
*Spongia venosa* Lamarck, 1814 (genus transfer & junior synonym)
*Spongia ventilabra* *Linnaeus, 1767* (genus transfer & incorrect spelling)
*Spongia ventilabrum* *Linnaeus, 1767* (genus transfer)

**Material examined**
  CFM-IEOMA-7376/i822_1, St 39, EB, ROV.

**Ecology notes**
  The single specimen was collected on a rhodolith bed in the summit of the EB at 132 m deep (Fig. 2C) where, according to preliminary analysis of ROV videos, it seems to be a rare species.

**Remarks**
  This is the first report of the species at the Balearic Islands. The species has been widely reported in the North Atlantic (*e.g. Alvarez & Hooper, 2002*), to Greenland (*Lundbeck, 1909*; *Hentschel, 1929*) and Canada (*Lambe, 1900*). In the Mediterranean, it has been reported northern of the Iberian Peninsula (*Uriz, 1984*), in the Alboran Sea (*Maldonado, 1992*) and Corsica (*Vacelet, 1961*).

### *Phakellia hirondellei* *Topsent, 1890*
**Synonymised names**
*Axinella hirondellei* *Topsent, 1890* (reverted genus transfer)
*Phakellia robusta var. Hirondellei* (*Topsent, 1890*) (status change)
*Tragosia hirondellei* (*Topsent, 1890*) (reverted genus transfer)

**Material examined**
  CFM-IEOMA-7377/i353, St 13, EB, BT; CFM-IEOMA-7378/i623, St 33, AM, RD.

**Ecology notes**
  The species was found at two stations of similar depth (135–147 m) in AM and EB. Both stations are located at the border of the summit, an area that may be affected by enhanced water current and an increase in nutrient and food supply (*Samadi, Schlacher & De Forges, 2007*; *Rogers, 2018*). This could explain the common presence of large erect

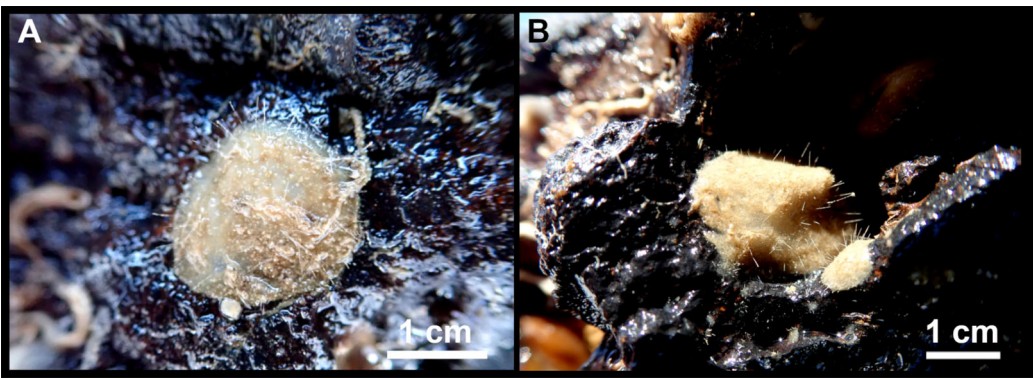

**Figure 9 *Heteroxya* cf. *beauforti*.** (A) Habitus of CFM-IEOMA-7380/i726 in fresh state. (B) Habitus of CFM-IEOMA-7382/i461 in fresh state (large patch).

sponges such as *Poecillastra compressa* (*Bowerbank, 1866*) on stations located at these areas (personal observations).

**Remarks**

The species is reported for the first time in the Mallorca Channel, being its second record at the Balearic Islands, where it was previously recorded by *Santín et al. (2018)* from the Menorca Channel. In the Mediterranean Sea, it is also known in the north of the Balearic Sea (*Uriz, 1984*) and in the Gulf of Lions, the Ligurian Sea and Corsica (*Fourt et al., 2017*) and the Alboran Sea (*Boury-Esnault, Pansini & Uriz, 1994*).

Family HETEROXYIDAE Dendy, 1905
Genus *Heteroxya* *Topsent, 1898*
***Heteroxya* cf. *beauforti***
(Figs. 9, 10 and 11; Table 4)

**Material examined**

CFM-IEOMA-7381/i444, St 27, SO, RD; CFM-IEOMA-7382/i461, St 28, SO, RD; CFM-IEOMA-7450/i487, St 30, SO, RD; CFM-IEOMA-7380/i726, St 35, EB, RD; CFM-IEOMA-7379/i727, St 35, EB, RD.

**Description**

Small encrusting patches, circular or irregular, up to 2 cm in diameter (Figs. 9A and 9B). Body less than 1 mm thick. Consistency hard and slightly flexible. Hispidation visible to the naked eye. Greyish in life and after preservation in EtOH. No pores observed.

**Skeleton**

A basal spongin layer adheres to the substrate and allows the whole body to be peeled-off with a scalpel. Just upon this layer there are Oxea II running parallel to the substrate. The choanosome has low spicule content. Choanosomal chambers are relatively well developed in the thicker parts of the sponge (Figs. 10A and 10B). Thick areas also have ascending tracts of Oxea II, with Oxea II placed in between. The choanosomal tracts are not present in the thinner areas (Fig. 10C). The basal layer and the choanosome have

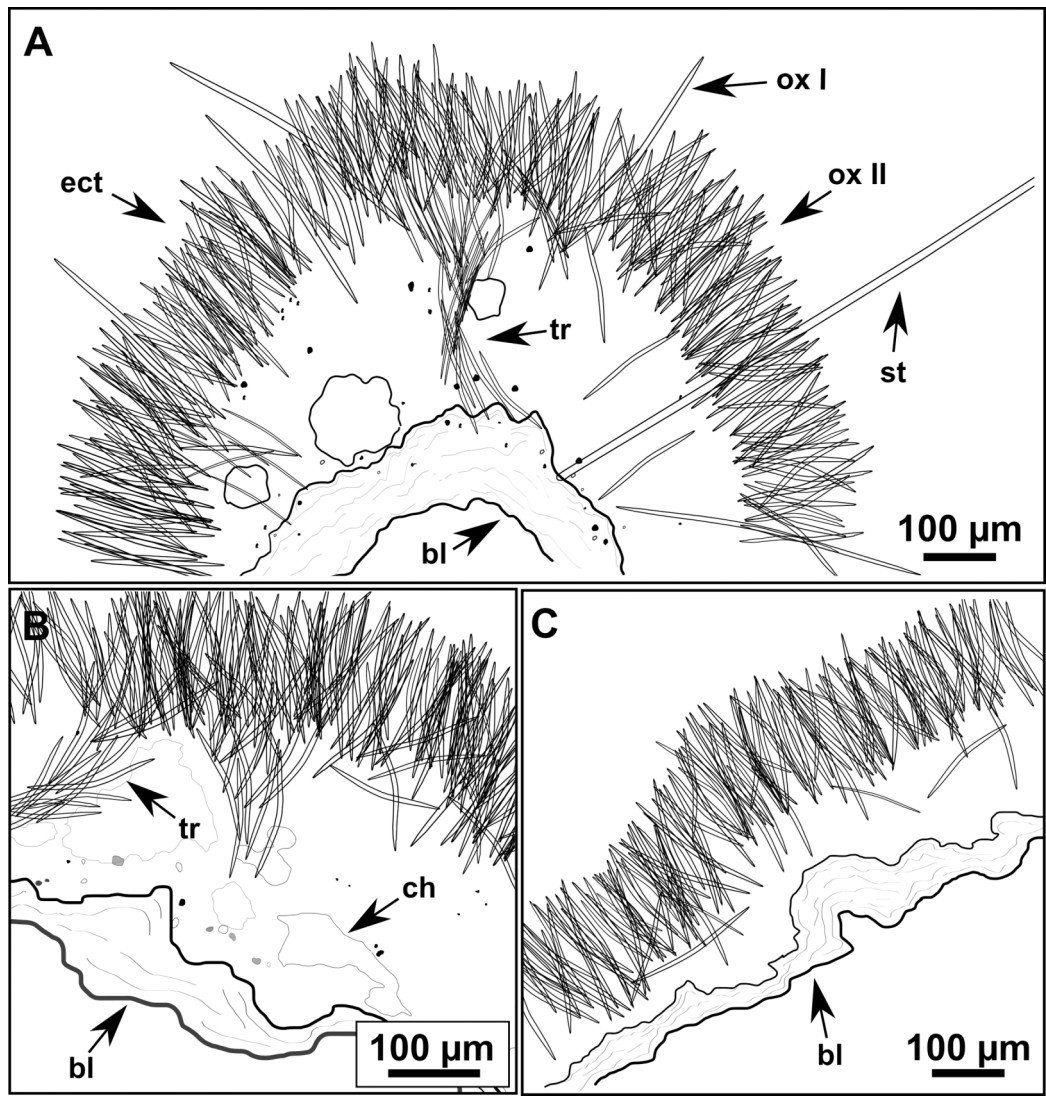

**Figure 10 Schematic illustration of *Heteroxya* cf. beauforti skeleton in transversal section.** (A) General view. (B) Body arrangement on a thick area. (C) Body arrangement on a thin area. (ox I) oxea I. (ox II) oxea II. (bl) basal lamina. (ect) ectosome. (ch) choanosome. (tr) spicule tracks.

abundant circular bodies 3-9 μm in diameter, dark or transparent (Fig. 10B). The ectosome is constructed by a dense palisade of Oxea II, perpendicular to the surface, with Oxea I placed in the same perpendicular position, emerging towards the exterior. Long styles are found here and there outcrossing the ectosome and causing the hispidation.

**Spicules**

Oxeas I (Fig. 11A): may be gently curved or bent in the middle, with sharp tips. They measure 319–482–623 × 7–10–15 μm.

Oxeas II (Figs. 11B): gently curved, curved or bent in the middle. Some stylote modifications present. Many with teratogenic parts like bifid tips, swellings or poliaxonal modifications (Figs. 11C and 11D). They measure 104–198–293 × 3–7–10 μm.

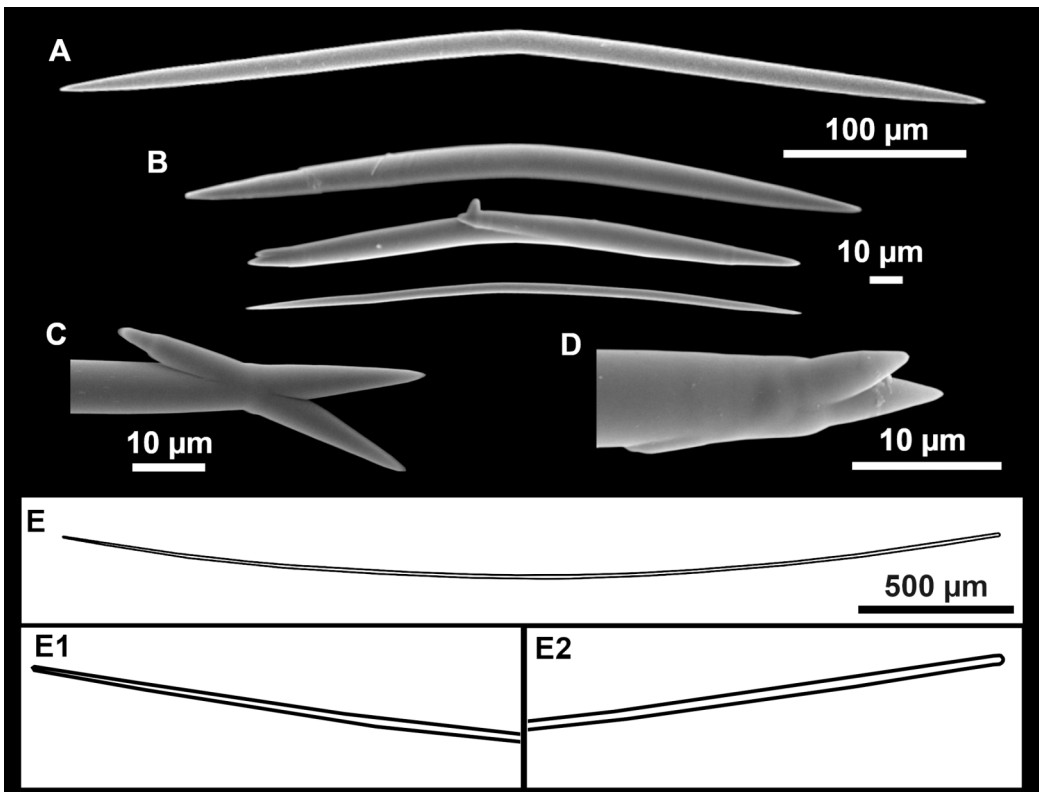

**Figure 11 Spicules of *Heteroxya* cf. *beauforti*.** (A) Large oxeas I. (B) Small oxeas II. (C–D) Detail of polyactinal teratogenic modifications of oxeas II. (E) Drawing of a style with details of the tip (E1) and the head (E2).                

Hispidating styles (Figs. 11E–11E2): very long and thin, curved, with round ends and sharp tips. Most broken, only three complete from specimen CFM-IEOMA-7381/i444, measuring 1151–3502 × 8–14 μm (*n* = 3).

**Ecological notes**

The species has been collected on smooth basaltic rocks between 270 and 325 m deep at SO and EB, where it seems to be rather common. Mostly associated with other minute encrusting sponges like *Hamacantha* spp. or *Bubaris* spp.

**Genetics**

Sequences of *COI* Folmer fragment and the *28S* C1-D2 domains were obtained from the specimen CFM-IEOMA-7380/i726. Both sequences were deposited at the Genbank, under the accession numbers MW858350 and MW881150, respectively.

**Remarks**

The genus *Heteroxya* contains two species, *H. corticata* Topsent, 1898 and *H. beauforti* Morrow, 2019. *Heteroxya corticata* is the type of the genus, known only from deep waters (1,200–1,600 m) of the Azores Archipelago. The species has two categories of oxeas, both microspined, and lacks styles. Conversely, *H. beauforti* is known from slightly shallower waters of Ireland (630–1,470 m), has smooth oxeas and posses long hispidation

**Table 4 Comparative characters of species of the genus *Heteroxya*.**

| Specimen | Oxea I | Oxea II | Style | Depth | Area |
|---|---|---|---|---|---|
| *Heteroxya corticata* | | | | | |
| *Topsent, (1898)* Syntypes redescribed by *Morrow et al. (2019)* | 1600–1700–2000 × 26–32–37, Microspined ends | 235–310–420 × 12–23 Pronounced spination (more at the tips) | np | 1165–1240 | Azores |
| *Heteroxya beauforti* | | | | | |
| *Morrow et al. (2019)* Holotype | 622–1030–1385 × 10–16–21 Smooth | 207–280–370 × 11–14–16 Smooth | 5000–5650–6300 × 23–25–27 | 629–1469 | Celtic Seas |
| *Heteroxya* cf. *beauforti* | | | | | |
| CFM-IEOMA-7380/i726 | 434–569 × 7–13 (*n* = 7) Smooth | 107–180–287 × 4–6–9 Smooth | Broken | 280–306 | EB St 35 |
| CFM-IEOMA-7381/i444 | 319–467–580 × 6–10–14 (*n* = 23) Smooth | 104–171–257 × 4–6–8 Smooth (*n* = 23) | 1151–3502/8–14 (*n* = 3) | 288–318 | SO St 27 |
| CFM-IEOMA-7382/i461 | 327–460–586 × 6–10–15 Smooth | 167–233–286 × 3–7–9 Smooth | Broken | 255–325 | SO St 28 |
| CFM-IEOMA-7379/i727 | 420–530–623 × 9–12–15 (*n* = 18) Smooth | 142–192–293 × 6–8–10 Smooth | Broken | 280–306 | EB St 35 |
| CFM-IEOMA-7450 /i487 | nm | nm | nm | 270–325 | SO St 30 |

**Note:**
Depth (m), area (SO, Ses Olives; EB, Emile Baudot) and sampling station (St; see R*study* in Table 1) where these specimens were collected are also shown. Spicule measures are given as minimum-mean-maximum for total length × minimum-mean-maximum for total width. A minimum of 30 spicules per spicule kind are measured, otherwise it is stated. All measurements are expressed in µm. Specimen codes are the reference numbers of the CFM-IEOMA/author collection. np, not present; nm, not measured.

styles (Table 4). The genus was reviewed by *Morrow et al. (2019)*, that sequence the *COI* of both holotypes. They found no differences between the *COI* of *H. corticata* and *H. beauforti* but conclude that morphological differences were enough to consider both as different species.

Morphologically, our material is more related to *H. beauforti* due to the abscense of microspined oxeas and the presence of hispidation styles. We have found circular bodies embedded in the choanosome and the basal layer, which can be equivalent to the spherulous cells found in *H. beauforti* (*Morrow et al., 2019*). However, oxea I, oxea II and styles are markedly shorter and thinner in our material than those of *H. beauforti*. Those differences may be a result of depth, nutrient, or temperature differences. On the other hand, the *COI* sequence of our material is identical to the sequences of *H. corticata* and *H. beauforti*. We have sequenced the *28S* C1-D2 domains, but there are no published sequences to compare. Considering the lack of genetic differences and the affinity of our material to *H. beauforti*, here we believe that erecting a new species is not justified. Future works using other markers will clarify if *H.* cf. *beauforti* and *H. beauforti* are conspecific, or if *H.* cf. *beauforti* is a different species.

*Heteroxya* cf. *beauforti* represents the first record of a species belonging to the genus *Heteroxya* in the Mediterranean Sea.

Family STELLIGERIDAE Lendenfeld, 1898
Genus *Paratimea* Hallmann, 1917

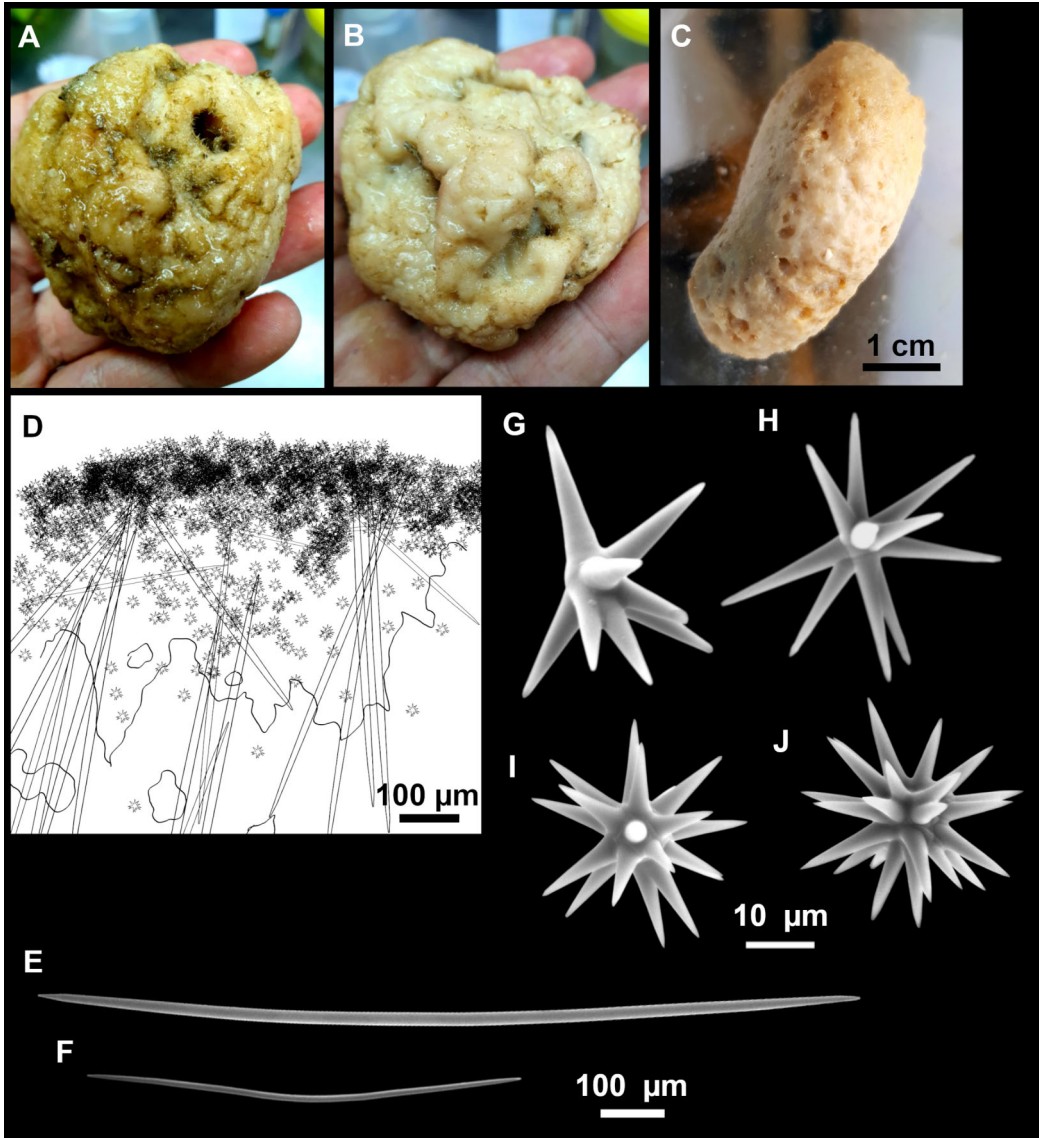

**Figure 12** *Paratimea massutii* **sp. nov.** (A–B) Habitus of the holotype CFM-IEOMA-7383/i403 in fresh state, on its upper (A) and lower (B) sides. (C) Habitus of the paratype CFM-IEOMA-7384/i420 preserved in EtOH. (D) Schematic illustration of a transversal section of the holotype. (E–J) SEM images of the Holotype. (E) Oxea I, (F) Oxea I (auxiliar spicule). (G–J) Oxyasters (all with same bar scale).

### *Paratimea massutii* sp. nov.

(Fig. 12; Table 5)

### Diagnosis

Massive ovoid sponge with oxeas as megascleres and oxeas as auxiliary spicules. Centrotylotism occasionally present in both. Oxyasters smooth.

### Etimology

Dedicated to Professor Enric Massutí, for his contribution to the knowledge of the benthic communities of the Balearic Islands.

**Table 5 Comparative characters of *Paratimea* spp. from the Mediterranean and the north-eastern Atlantic, including *Paratimea massutii* sp. nov.**

| Species/specimen | Megascleres | Accessory oxeas | Oxyaster | Other spicules | External morphology | Depth | Area |
|---|---|---|---|---|---|---|---|
| *Paratimea massutii* **sp. nov.** | | | | | | | |
| CFM-IEOMA-7383/ i403 Holotype | Oxeas 910–1419–1711 × 16–24–33 (n = 17) | 469–681–827 × 3–8–10 (n = 7) | Smooth, 25–36–45 9–25 rays | np | Massive, lobate surface, whitish with diatom brownish on the upper side | 151 m | EB St 15 |
| CFM-IEOMA-7384/ i420 Paratype | Oxeas 1130–1374–1561 × 11–20–28 | 556–755–862 × 3–6–8 | Smooth, 27–39–57 7–20 rays (occasionally 2 rays) | np | same as i403 | 156 m | EB St 17 |
| *Paratimea oxeata* Pulitzer-Finally, (1978) | | | | | | | |
| Holotype | 1000–1450 × 14–24 | 250–650 × 3–7 | 40–60 | np | Thickly encrusting, up to 4 × 5 × 0,4 cm, drab color in life, white after formalin and EtOH | 60 and 100–110 m | Bay of Naples |
| Bertolino et al. (2013) | 810–961–1200 × 15–18–25 | 300–547–700 × 3–5–5 | 25–42–60 | np | Very small (0.5 cm²) insinuating sponge, grey colored in dry state. | 35 m | Ligurian Sea |
| Morrow et al. (2019) | 1000–1500 × 14–24 | 250–650 × 3–7 | 20–40 but up to 60 when reduced rays 4–12 rays | np | Massive lobose, surface conulose, oscules arranged on top of raised humps, Pale yellow-cream | Caves, 15–20 m | Gulf of Lion |
| *Paratimea loricata* (Sarà, 1958a) | | | | | | | |
| Holotype | Oxeas, poliaxonic and aberrant terminations. Mostly non-centrotylota. 320–420 × 5–7 (most common) and 600 × 15 (n = 1) | Centrotylote 105–180 × 2–3 | Large: 40–50 Small (uncommon): 12–20 | Tylostyles, trilobated head 130–170 × 4–7 | Encrusting, elastic but friable, whitish-yellow after EtOH | Not specified, infralittoral | Ligurian Sea |
| *Paratimea pierantonii* (Sarà, 1958b) | | | | | | | |
| Holotype and paratypes | Styles and Subtylostyles: 1530–2550/12–18 | 650–1175 × 4–10, centrocurved, non-centrotylote | 15–25 | np | Cushion shaped with papillae. Hispid, smooth to the touch. Orange yellow at the surface, brownish inside. | 30 cm, tidal cave | Tyrrhenian Sea |

(Continued)

| Species/specimen | Megascleres | Accessory oxeas | Oxyaster | Other spicules | External morphology | Depth | Area |
|---|---|---|---|---|---|---|---|
| *Paratimea arbuscula (Topsent, 1928)* | | | | | | | |
| Holotype | Curved or flexuous, centrotylote. Some modified to styles. 560–1000 × 5–12 | nr | Without centrum, with conical, acanthose actines, 15–60 most with 12 rays | np | Small arbuscular sponge, up to 1 cm in heigth 1 mm in width, hispid. Whitish. Asters concentrated at the periphery | 650–914 m | Azores |
| *Paratimea duplex (Topsent, 1927)* | | | | | | | |
| Reproduced from the redescription in *Morrow et al. (2019)* | Centrotylote oxeas 2000–2600 × 20–40, styles to subtylostyles 1600–1800 × 25–35 | Weakly centrotylote 360–770 × 7–9 | Without centrum, smooth rayed, 50–100 10–15 rays | np | cushion shaped, 3 mm thick, with a conulose surface | 240–2165 m | North Atlantic Ocean |
| *Paratimea constellata (Topsent, 1904)* | | | | | | | |
| Holotype, reproduced from *Morrow et al. (2019)* | Long, slender tylostyles 2500–3000 × 13–14 | Centrotylote oxeas 379–670–900 × 8–10 | Smooth-rayed euasters 14–30–46 | np | Cushion shaped, 2–3 mm thick, yellow gold | 40 m | Roscoff, Celtic seas |
| *Paratimea loenbergi (Alander, 1942)* | | | | | | | |
| Reproduced from the redescription of the Holotype in *Morrow et al. (2019)* | 1350–3000 × 10–13–15 (n = 4); head, 16–20–27 | Slightly bent, 530–712–930 × 5–5–6 (n = 7) | Smooth 22–28–36 | Small category of tylostyles not found by *Morrow et al. (2019)* but mentioned in the original description, measuring 180–225 × 12–15 | Thin, hispid crust, pale yellow. | 60 m | Väderöfjord, Sweden |
| *Paratimea hoffmannae Morrow & Cárdenas, 2019* | | | | | | | |
| Holotype, original description | Large, curved oxeas, occasionally centrotylote 2056–2187–2250 × 25–26–28 | Rare, bent, occasionally centrotyle 353–446–520 × 3–4–5 | Asymmetic 42–60–81 μm 7–18 smooth, tapering rays | np | Massive, subspherical. Holotype is ~7 in diameter. Surface covered in large conules, 1–4 mm in height. Creamish white. | 328 m (Holotype) 1500 m (Paratype) | Norway (Holotype) Ireland (Paratype) |

**Note:**

Depth, area (EB, Emile Baudot) and sampling station (St; see R*study* in Table 1) where these specimens were collected are also shown. Spicule measures are given as minimum-mean-maximum for total length × minimum-mean-maximum for total width. A minimum of 30 spicules per spicule kind are measured, otherwise it is stated. All measurements are expressed in μm. Specimen codes are the reference numbers of the CFM-IEOMA/author collection. np, not present; nr, not reported.

**Material examined**
*Holotyope:* CFM-IEOMA-7383/i403, St 15, EB, BT.
*Paratype:* CFM-IEOMA-7384/i420, St 17, EB, BT.

**Description**
   Both specimens are massive, subspherical, the largest (holotype, CFM-IEOMA-7383/i403; Fig. 12A) measuring about 5 cm in diameter, having a lobose surface with grooves and humps. Skin of a leathery touch, hispid only in the grooves. Color in life differing between the upper and the lower faces, the former having the first a brownish tinge while the latter a whitish to beige shade (Figs. 12A and 12B). After preservation in EtOH the whole body turns homogeneous vanilla cream (Fig. 12C). Both specimens have 4–6 circular oscula, 1–2 mm in diameter, scattered throughout the body. However, the holotype also has a main large and circular osculum, about 1 cm in diameter, on the upper side. Both specimens expelled a considerable amount of mucus when collected.

**Skeleton**
   Ectosome not separable from the choanosome, formed by a dense crust of oxyasters and tangential principal and auxiliary oxeas. Choanosome composed of irregularly arranged oxeas and oxyasters, although radial bundles of large oxeas are present in the periphery, supporting the ectosome (Fig. 12D).

**Spicules**
*Megascleres*
Oxea I (Fig. 12E): robust and fusiform, some double bent, sometimes slightly centrotylote. They measure 910–1390–1711 × 11–21–33 µm.
*Auxiliary spicules*
Oxea II (Fig. 12F): uncommon. Bent or slightly sinuous, sometimes centrotylote. They measure 469–746–1088 × 3–7–10 µm.
*Microscleres*
Oxyasters (Figs. 12G–12J): with long, smooth and sharp rays. About 7–25 rays, occasionally less. Smaller ones tend to have more rays than larger ones, measuring 25–38–57 µm. Occasionally, some two-rayed oxyaster present.

**Ecological notes**
   Found at two stations on calcareous gravel bottoms on the summit of EB (155 and 167 m deep), which was dominated by sponges such as *Hexadella* sp., *Phakellia robusta* and different species of the order Tetractinellida. A large number of the brachiopod *Gryphus vitreus* (*Born, 1778*) and echinoderms were also recorded.

**Genetics**
   Sequences of *COI* Folmer fragment was obtained from the Holotype (CFM-IEOMA-7383/i403) and deposited at Genbank under the accession number MW858351.

**Remarks** (see Table 5 for a detailed comparison with other *Paratimea* spp.)

Morphologically, the species resembles *Paratimea oxeata Pulitzer-Finali, Hadromerida & Poecilosclerida, 1978*, a Mediterranean species reported at rocky and muddy bottoms, at 35–60 and 110 m deep, respectively (*Pulitzer-Finali, Hadromerida & Poecilosclerida, 1978*; Bertolino et al., 2013), and at submarine caves at 15–20 m deep (*Morrow et al., 2019*). However, *P. massutii* **sp. nov.** is massive, a feature only shared with the cave specimen (S153) reported by *Morrow et al. (2019)*. Notwithstanding, in *P. massutii* **sp. nov.** oxeas I are thicker, oxeas II longer and oxyasters slightly larger and with more actines (2–25 *vs.* 4–12). A comparison of the *COI* sequences between the holotype of *P. massutii* **sp. nov.** and the cave specimen confirms those morphological differences, with 15 bp differences and a *p*-distance of 2%. On the other side, both the holotype and the specimens studied by *Bertolino et al. (2013)* differ from *P. massutii* **sp. nov.** in being cushion shaped or encrusting and having smaller oxeas. Unfortunately, no sequences of *Bertolino et al. (2013)* specimens are available to compare. *Paratimea massutii* **sp. nov.** is also similar to *P. hoffmannae Morrow et al. (2019)*, a North Atlantic species found in Norway and Ireland that is also massive and subspherical and has oxeas as both megascleres and auxiliary spicules. However, the large oxeas are much larger and thicker than in *P. massutii* **sp. nov.**, in contrast to the auxiliary spicules, which are shorther and thinner. Also, the oxyasters of *P. hoffmannae* are larger and with less actines. As for *P. oxeata*, *COI* sequences between *P. hoffmannae* and *P. massutii* **sp. nov.** are notably distant, with 13 bp differences and a *p*-distance of 2%. A similar case happens with *P. lalori* Morrow, 2019 from Ireland. This species is also massive-subspherical with oxeas as main megascleres and auxiliary spicules. Just as in *P. hoffmannae*, megascleres of *P. lalori* are longer and thicker than those of *P. massutii* **sp. nov.**, auxiliary spicules are shorter and thinner and oxyasters slightly larger and with fewer actines.

*Paratimea massutii* **sp. nov.** also differs from the other Mediterranean *Paratimea* spp. as follows: *P. loricata* (*Sarà, 1958a*) is encrusting, has much smaller oxeas I and oxeas II and two categories of oyasters, and bears tylostyles; *P. pierantonii* (*Sarà, 1958b*) is cushion-shaped, has styles and subtylotyles as megascleres, longer, thicker, and never centrotylote oxeas II and smaller oxyasters.

Also, *P. massutii* **sp. nov.** differs from North-eastern Atlantic *Paratimea* spp. as follows: *P. constellata* is cushion shaped, has tylostyles and smaller oxyasters; *P. arbuscula* (*Topsent, 1928*), is arbustive, lacks auxiliary spicules and has smaller, acanthose oxyasters; *Paratimea duplex* (*Topsent, 1927*) is cushion shaped, has much larger oxeas I, bears styles, subtylostyles, and two categories of oxyasters; *P. loennbergi* (*Alander, 1942*) is thinly encrusting, has tylostyles and smaller oxyasters.

This is the first report of the genus *Paratimea* in the Balearic Islands, and the deepest record in the Mediterranean Sea.

Order BUBARIDA Morrow & Cárdenas, 2015
Family BUBARIDAE Topsent, 1894
Genus *Rhabdobaris Pulitzer-Finali, 1983*
***Rhabdobaris implicata** Pulitzer-Finali, 1983*

**Synonymised names**

*Cerbaris implicatus* (*Pulitzer-Finali, 1983*)

**Material examined**

CFM-IEOMA-7385/i338_2_1, St 11, EB, BT; CFM-IEOMA-7386/i698, St 34, EB, RD.

**Ecological notes**

Uncommon sponge found at two stations on the EB summit at 117 and 152 m deep, growing on living rhodoliths. Both stations were rich in massive demosponges, including large Tetractinellids, *Petrosia* (*Petrosia*) *ficiformis* and *P.* (*Strongylophora*) *vansoesti*.

**Remarks**

This is the third time that the species is recorded, previously only known from the holotype collected in Corsica (*Pulitzer-Finali, 1983*) and the neotype collected at the Alboran Island (*Sitjà & Maldonado, 2014*).

Order DESMACELLIDA Morrow & Cárdenas, 2015
Family DESMACELLIDAE *Ridley & Dendy, 1886*
Genus *Dragmatella* Hallman, 1917
**Dragmatella aberrans** (*Topsent, 1890*)
(Fig. 13; Table 6)

**Material examined**

CFM-IEOMA-7387/i52_b1, St 2, SO, BT; CFM-IEOMA-7388/i175, St 5, EB, BT.

**Description** (modified from *Hooper & Van Soest, 2002*)

Small hollow sponge encrusting on stones or corals. Up to 2 cm in diameter. Whitish grey in life and after preservation in EtOH. Surface smooth, but provided with long thin, pointed fistules (Figs. 13A and 13B).

**Skeleton**

Ectosome composed of parallel tight tracts of styles, disposed in 4–5 layers of 30–50 μm in thickness (Fig. 13C). The raphides, sometimes grouped in trichodragmata, are scattered in the ectosomal and choanosomal tracts. Choanosome is cavernous (Fig. 13D), with tracts of styles, about 200 μm long, verging from a basal layer towards the ectosome.

**Spicules**

*Megascleres*

Styles (Figs. 13E and 13E1) fusiform, tappering towards the head, slightly or abruptly bent. They measure 349–546–676 × 6–10–15 μm.

*Microscleres*

Raphides (Figs. 13F and 13F1) abundant, straight, with an irregular shaft and one end hook-shaped, occasionally with central swellings. They measure 162–195–222 × 1–2–3 μm.

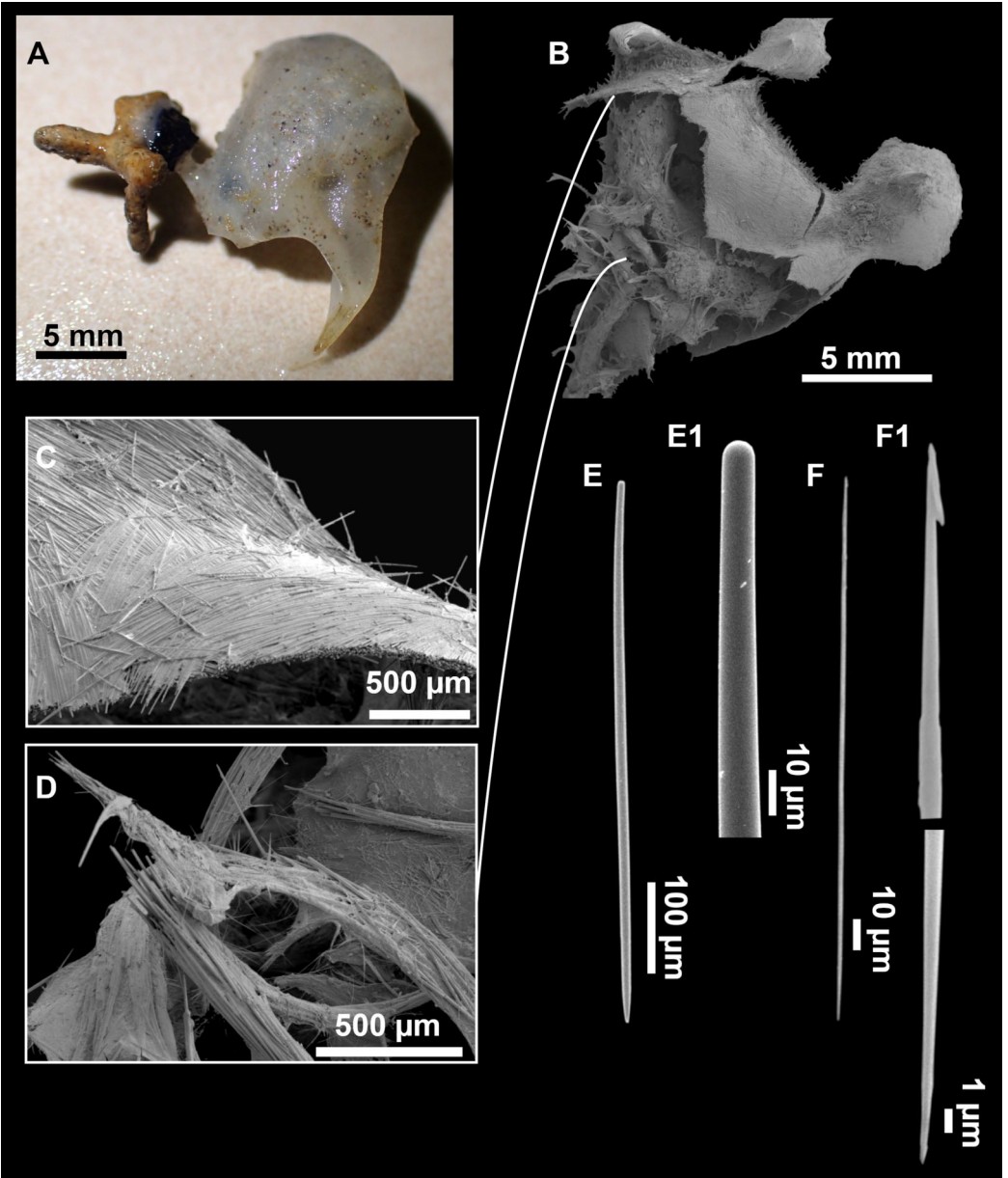

**Figure 13 _Dragmatella aberrans_ (_Topsent, 1890_).** (A) Habitus of CFM-IEOMA-7388/i175 preserved in EtOH. (B–D) SEM images of the skeletal structure of CFM-IEOMA-7388/i175. (B) General view of the skeletal arrangement. (C) Detail of the ectosome. (D) View of the ascending choanosomal tylostyle tracks. (E–E1) Mycalostyles. (F) Raphides with (F1) Detail of the hook-shaped ends and central irregularities.

## Ecological notes

Abundant species on sedimentary bottoms, with rests of calcareous shells and corals, found in SO, AM and EB and, to a lesser extent, on trawl fishing grounds of the continental shelf off Mallorca (between 138 and 362 m deep). On the same bottoms other small encrusting sponges such as _Hamacantha_ spp. or _Bubaris_ spp., the pedunculated

**Table 6 Comparative characters of representative reports of *Dragmatella aberrans*.**

| Specimen | Styles | Raphides | Depth | Area |
|---|---|---|---|---|
| *Topsent (1892)* | 600 | 180 | 135–134 | Cantabric Sea |
| *Topsent (1928)* | 600–800 × 9–11.5 | 70–200 × 12–20 | 552–1262 | Cap Sines (Portugal) |
| *Vacelet (1969)* | 350–600 × 6–13 | 150–210 | 250–324 | Cassidaigne (Gulf of Lion) |
| *Pulitzer-Finali (1983)* | 400–600 × 6–14 | 200 | 128–150 | Off Calvi (Corsica) |
| *Boury-Esnault, Pansini & Uriz, 1994* | 315–571–631 × 5–11–16 | 95–207–260 × 0.4–2–3 | 485 (Atlantic) 195 (Mediterranean) | Atlantic and Alboran Sea |
| CFM-IEOMA-7387/i52_b1 This work | 349–555–676 × 6–9–13 | 162–197–222 × 1–2–3 | 275 | SO St 2 |
| CFM-IEOMA-7388/i175 This work | 351–539–651 × 8–11–15 | 163–193–214 × 1–2–3 | 138 | EB St 5 |

Note:
Depth (m), area (SO, Ses Olives; EB, Emile Baudot) and sampling station (St; see R*study* in Table 1) where these specimens were collected are also shown. Spicule measures are given as minimum-mean-maximum for total length × minimum-mean-maximum for total width. A minimum of 30 spicules per spicule kind are measured, otherwise it is stated. All measurements are expressed in μm. Specimen codes are the reference numbers of the CFM-IEOMA/author collection.

*Rhizaxinella pyrifera* (*Delle Chiaje, 1828*) and *Thenea muricata* (*Bowerbank, 1858*), the brachiopod *Gryphus vitreus* (*Born, 1778*) and small crustaceans are to be found.

**Remarks**

The species is easily distinguished by its hollow body and the possession of both styles and raphides. The latter have singular hook-shaped ends, a feature that had not been recorded before, and that is similar to the raphides found in some species of the genus *Dragmaxia* (Order Axinellida) (*Hooper & Van Soest, 2002*). No molecular data are available for *Dragmatella*, but a phylogenetic relationship with *Dragmaxia* is unlikely, given the possession of styles and the skeletal arrangement of both genera. Therefore, hook-shaped raphide are probably homoplasic.

This is the first report of the species in the Balearic Islands. In the Mediterranean Sea it has been recorded at the Gulf of Lions (*Vacelet, 1969*), Corsica (*Pulitzer-Finali, 1983*) and the Alboran Sea (*Boury-Esnault, Pansini & Uriz, 1994*; *Sitjà & Maldonado, 2014*). In the North Atlantic Ocean, this species has been recorded at several locations, including the coast of Portugal (*Topsent, 1895*), the Josephine Bank (*Topsent, 1928*) and the Cantabric Sea (*Topsent, 1890*).

Order HAPLOSCLERIDA *Topsent, 1928*
Family CHALINIDAE Gray, 1867
Genus *Haliclona* Grant, 1841
Subgenus *Soestella* De Weerdt, 2000
***Haliclona* (*Soestella*) *fimbriata* Bertolino & Pansini, 2015**

**Material examined**

CFM-IEOMA-7389/i825_1, St 40, EB, ROV.

**Ecological notes**

The species was spotted regularly at the rhodolith beds of the EB summit, between 134 and 150 m deep. However, it was less abundant and not forming patches, as occurs in

some areas of the Gulf of St. Eufemia in the Tyrrhenian Sea, where *Bertolino et al. (2015)* reported densities of $7.4 \pm 0.7$ specimens/m$^2$.

This is the second report of the species, previously recorded only at the Gulf of St. Eufemia (southern Tyrrhenian Sea; *Bertolino et al., 2015*), expanding its distribution range towards the westernmost part of the Mediterranean Sea.

Family PETROSIIDAE Van Soest, 1980
Genus *Petrosia* Vosmaer, 1885
Subgenus *Strongylophora* Dendy, 1905
***Petrosia* (*Strongylophora*) *vansoesti* Boury-Esnault, Pansini & Uriz, 1994**

### Material examined

CFM-IEOMA-7390/i192_A and CFM-IEOMA-7391/i192_B, St 6, EB, BT; CFM-IEOMA-7392/i313_P and CFM-IEOMA-7393/i313_G, St 11, EB, BT; CFM-IEOMA-7394/i351, St 13, EB, BT; CFM-IEOMA-7395/i694, St 34, EB, RD.

### Ecological notes

Large amounts of *P.* (*S*) *vansoesti* were collected from various stations in the summit of the EB, suggesting that it is an important species inhabiting Mediterranean seamounts and probably a habitat builder that confers three-dimensionality to the seafloor.
The species was found from 116 to 152 m deep, on stations with living and dead rhodoliths and gravels, associated with large sponges such as *P.* (*P.*) *ficiformis* and some tetractinellids and halichondrids. Many groups of invertebrates, such as small crustaceans and echinoderms, were also observed at these stations.

### Remarks

This is the first record of the species in the western Mediterranean. The type locality is the Gulf of Cadiz, in the north-eastern Atlantic. In the Mediterranean it has been recorded in marine caves at both the Ionian Sea (*Costa et al., 2019*) and the Aegean Sea (*Gerovasileiou & Voultsiadou, 2012*). It has been also recorded at the Levantine Sea, living on rocks at depths shallower than 3 m (*Evcen & Çinar, 2012*). On the Balearic Islands, the species has only been collected in EB.

Subgenus *Petrosia* Vosmaer, 1885
***Petrosia* (*Petrosia*) *raphida* Boury-Esnault, Pansini & Uriz, 1994**
(Fig. 14; Table 7)

### Material examined

CFM-IEOMA-7396/POR406, St 1, south-east of Menorca, GOC-73; CFM-IEOMA-7397/i178_3, St 5, EB, BT; CFM-IEOMA-7451/i242 and CFM-IEOMA-7398/i254_2, St 8, AM, BT; CFM-IEOMA-7399/i305, St 10, AM, BT; CFM-IEOMA-7400/i312, St 11, EB, BT.

### Description

Massive sponges, the largest collected specimen measuring about 4.5 cm in diameter and 2.5 cm in height (Fig. 14A). Whitish in life, beige after preservation in EtOH.

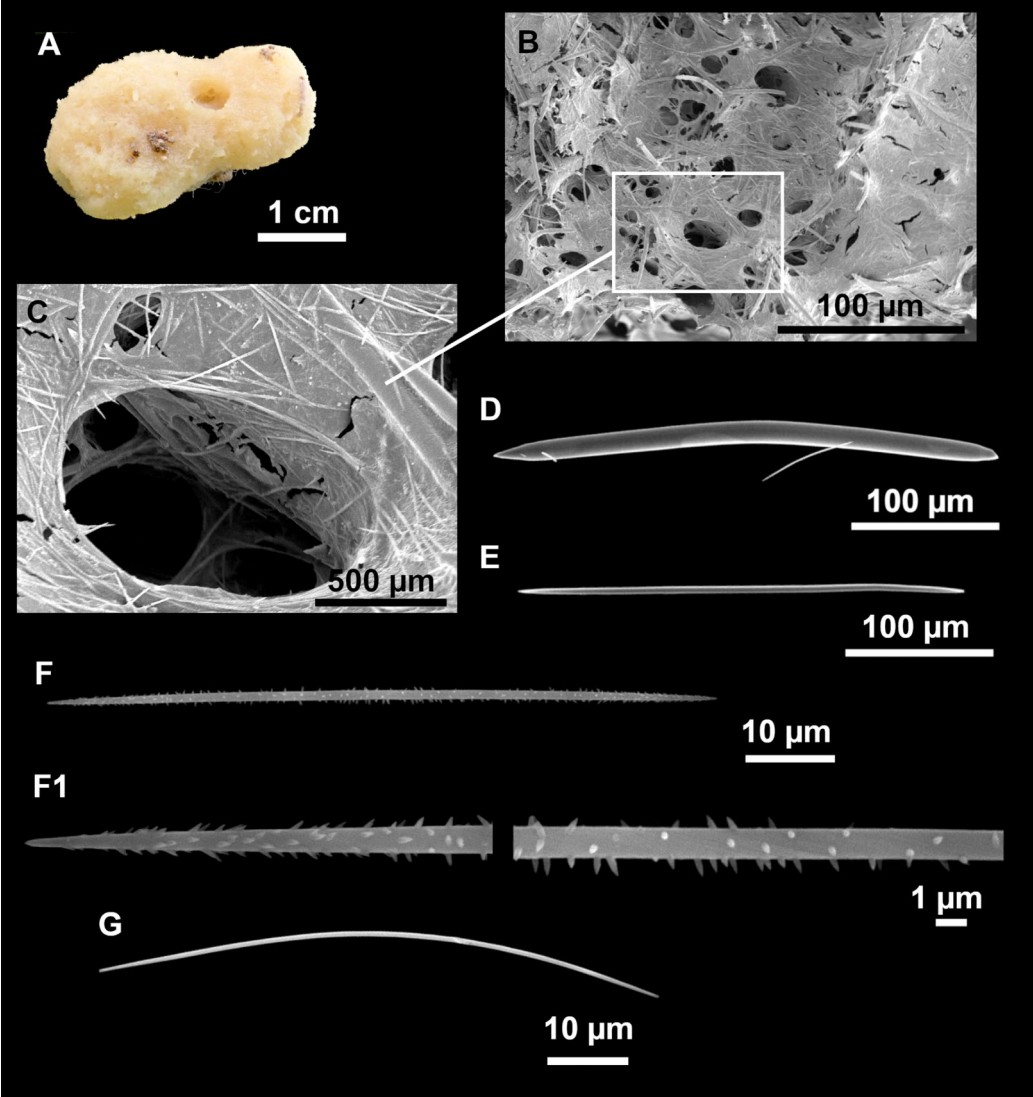

**Figure 14** *Petrosia (Petrosia) raphida Boury-Esnault, Pansini & Uriz, 1994.* (A) Habitus of CFM-IEOMA-7451/i242, preserved in EtOH. (B) SEM image of the choanosome. (C) Detail of a choanosomal chamber. (D) Oxeas. (E) Young stages of oxeas. (F–F1) Acanthoses raphides. (G) Smooth raphides.

Consistency hard, slightly crumbly. Surface rough due to minute conules, although in some specimens this is less obvious. There are 1 to 6 circular oscules of 2–5 mm diameter.

**Skeleton**

Ectosome forming a detachable crust not evident to the naked eye, tightly adhering to the choanosome, and made of irregular net of polygonal to triangular meshes. Meshes are constituted by one or two spicules. Spongin is present and fully embedded with raphides.

**Table 7 Comparative characters from published records of *Petrosia (Petrosia) raphida* *Boury-Esnault, Pansini & Uriz, 1994* and present work.**

| Specimen | Oxeas | Raphides | Depth | Area |
|---|---|---|---|---|
| *Boury-Esnault, Pansini & Uriz (1994)* Holotype | 354–449–499 × 26–32–36 (strongyles) | 81–95–108 × 1 | 580 | Gibraltar |
| *Sitjà et al. (2019)* | 290–500 × 20–25 (rarely as short as 7.5) | 75–100 × 1 (some without spines) | 530–573 | Volcano of Gulf of Cadiz (Pipoca) |
| CFM-IEOMA-7396/POR406 This work | 271–369–432 × 9–13–16 | 62–78–91 × 1–1–2 | 134 | South-east of Menorca St 1 |
| CFM-IEOMA-7397/i178_3 This work | 242–378–450 × 10–16–19 | 72–80–89 × 2–3–4 | 138 | EB St 5 |
| CFM-IEOMA-7451/i242 This work | 268–333–380 × 11–14–17 | 70–80–91 × 1–2–2 | 101 | AM St 8 |
| CFM-IEOMA-7398/i254_2 This work | 300–378–426 × 9–15–19 | 66–75–86 × 1–2–2 | 101 | AM St 8 |
| CFM-IEOMA-7399/i305 This work | 242–346–394 × 9–15–19 | 65–75–88 × 1–2–2 | 118 | AM St 10 |
| CFM-IEOMA-7400/i312_1 This work | 349–403–453 × 8–15–19 | 70–79–95 × 1–2–2 | 152 | EB St 11 |

**Note:**
Depth (m), area (SO, Ses Olives; AM, Ausias March; EB, Emile Baudot) and sampling station (St; see R*study* in Table 1) where these specimens were collected are also shown. Spicule measures are given as minimum-mean-maximum for total length × minimum-mean-maximum for total width. A minimum of 30 spicules per spicule kind are measured, otherwise it is stated. All measurements are expressed in μm. Specimen codes are the reference numbers of the CFM-IEOMA/author collection.

Choanosome (Figs. 14B and 14C) with an isotropic net of pauci-spicular spicule tracts covered by spongin, forming roundish meshes. These meshes are abundantly embedded by raphides. The tracts tend to condense towards the surface, supporting the ectosome.

**Spicules**

Oxeas (Figs. 14D and 14E): curved, with mucronated ends. Some polyaxonal modification in the shaft and ends may be present. They measure 242–372–450 × 9–15–19 μm, although underdeveloped stages (196–368/3–8 μm) are present. Styles and strongyles of the same length and width as the oxeas, present but scarce.

Raphides (Figs. 14F and 14F1): slightly curved, most minutely spined, although smooth ones are also present (Fig. 14G). They measure 62–77–95 × 1–2–2 μm.

**Ecological notes**

This species is very common in both AM and EB at the 101–152 m bathymetric range, and has been also found at the same depths off the southern coast of Menorca (Table 7). It can be found as a free-living sponge or growing attached to small fragments of calcareous sediments. However, it is also commonly found as an epibiont of other sponges and rhodoliths. The species seems to prefer massive specimens of *Hexadella* sp. and *Halichondria* sp. as substrate.

**Remarks**

The species is easily recognized due to the presence of characteristic spined raphides, added to other Petrosid features such as the skeletal architecture and the morphology of the oxeas. Remarkably, the specimens described in this study differ from the two

previous reports in having much smaller oxeas (see below in brackets). This could be explained by the scarcity of nutrients in waters around the Balearic Islands, the bathymetric range in which the specimens were collected and/or differences in water temperature, seasonal variability and population phenotypes (*Simpson, 1978*; *Valisano et al., 2012*). These variables could also be the cause of differences in the morphology of the megascleres already noted by *Sitjà et al. (2019)* when comparing their material with the holotype. In the specimens from the Gulf of Cadiz (north-eastern Atlantic), strongyles were rare. Instead, megascleres consisted mostly of oxeas with stepped tips and some occasional stylote or strongylote modifications. This last feature is shared with specimens of the Balearic Islands but not with the holotype, whose spicules have mostly strongylote extremities. These differences may be related to variations in nutrient regimes between the Balearic Islands and these areas (*Santinelli, 2015*).

This is the first record of the species in the Mediterranean, but at a considerably shallower depth (101–152 m) than in the north-eastern Atlantic, where the species was reported at 580 m deep in the Strait of Gibraltar (*Boury-Esnault, Pansini & Uriz, 1994*) and at 530–575 m deep in the Gulf of Cadiz (*Sitjà et al., 2019*) (Table 7).

Family PHLOEODICTYIDAE Carter, 1882
Genus *Calyx* Vosmaer, 1885
***Calyx* cf. *tufa***
(Fig. 15; Table 8)

**Material examined**

CFM-IEOMA-7403/i525, St 24, AM, BT; CFM-IEOMA-7401/i75, St 3, AM, BT; CFM-IEOMA-7402/i515, St 23, AM, BT.

**Description**

Large, massive and semicircular sponges, up to 15 cm in diameter and 5 cm in height (Fig. 15A). Surface smooth to the touch; consistency stony hard and uncompressible. Choanosome slightly friable and cavernous. Color in life beige, with pink tints in the upper side of the body and whitish beige in the lower. It became homogeneous brownish beige after preservation in EtOH. A total of Two to three large and circular oscula are located in the upper side of the body, measuring 1.3 cm in diameter. Ostia grouped in poral areas of the ectosome (Fig. 15B).

**Skeleton**

The ectosome (Fig. 15C) is formed by a crust of tangential spicules, forming triangular paucispicular meshes that become less dense at the poral areas. Spongin present but not abundant, with a granular appearance due to the presence of spherulous cells filled with granules (Fig 15C, arrow).

The choanosome (Figs. 15D and 15E) is mostly composed of a rather isotropic, unispicular net of spicules.

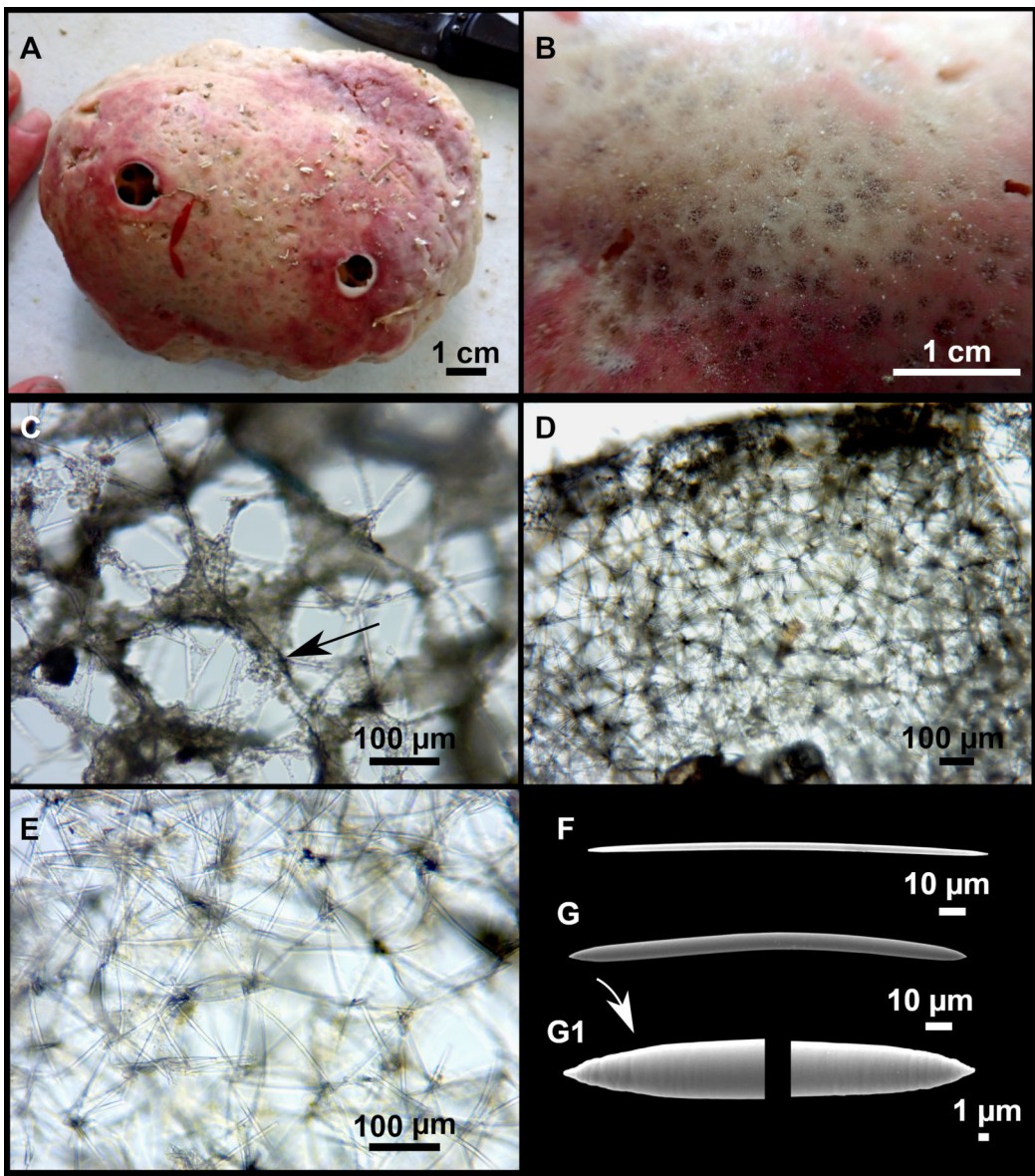

**Figure 15 *Calyx* cf. *tufa*.** (A) Habitus of CFM-IEOMA-7403/i525 in fresh state. (B) Detail of the ectosome with poral areas. (C) View of a poral area of the ectosome with spherulous cells (arrow). (D) Transversal section of the choanosome. (E) Detail of the reticulation of the choanosome. (F–G) Immature and mature oxeas, with (G1) Detail of the tips of (G).

### Spicules

Oxeas (Figs. 15F and 15G): slightly curved, with stepped or slightly mucronate points (Fig. 15G1) and rarely bent in the middle. They measure 132–173–206 × 4–7–9 μm.

### Genetics

Sequences of *COI* Folmer and *28S* C1–D2 domains were obtained from the specimen CFM-IEOMA-7403/i525 and deposited in Genbank under the accession numbers MW858349 and MW881149, respectively.

**Table 8 Comparative characters from *Calyx* cf. *tufa* and *Calyx tufa* Ridley & Dendy, 1886.**

| Specimen | Oxeas | External morphology | Depth | Area |
|---|---|---|---|---|
| *Calyx* cf. *tufa* | | | | |
| CFM-IEOMA-7403/i525 This work | 146–170–189 × 6–7–8 | Large, massive, roundish. Surface smooth. Stony hard and uncompressible. Ectosomal crust present. Beige with pink tints at the upper side. Whitish beige after EtOH | 114 | AM St 24 |
| CFM-IEOMA-7402/i515 This work | 140–171–205 × 4–7–9 | As the holotype | 113 | AM St 23 |
| CFM-IEOMA-7401/i75 This work | 132–178–206 × 4–6–9 | As the holotype | 105 | AM St 3 |
| *Calyx tufa* | | | | |
| *Ridley & Dendy (1886)* holotype | 200 × 10 | Massive, cake-like. Firm, almost stony, but brittle. Surface smooth but uneven. Dermal membrane (=ectosomal crust) readily peeling off. Vents rather small, circular, flush. Greyish yellow. | 219 | St Lago, Cape Verde |
| *Topsent (1892)* | nr | Firm but crumbly. Without ectosomal crust due to damaging. Light brown. | 300 | Cantabrian Sea |

Note:
Depth (m), area (AM, Ausias March) and sampling station (St; see R*study* in Table 1) where these specimens were collected are also shown. Spicule measures are given as minimum-mean-maximum for total length × minimum-mean-maximum for total width. A minimum of 30 spicules per spicule kind are measured, otherwise it is stated. All measurements are expressed in µm. Specimen codes are the reference numbers of the CFM-IEOMA/and author collection. nr, not reported.

### Ecological notes

The species was found only at the summit of AM between 105 and 114 m deep, associated with rhodolith beds. It has also been found amongst with diverse set of sponges, including large Tetractinellids and other sponges such as *Hexadella* sp., *Axinella* spp. or *P. (P.) raphida*, as well as among many other invertebrates typically inhabiting the rhodolith beds, like small crustaceans and echinoderms. The pink coloration of its upper skin is probably caused by symbiotic cyanobacteria, as commonly happens in other Haplosclerids (*Rützler, 1990*).

### Remarks

There are only two reported species of *Calyx* from the north-eastern Atlantic and the Mediterranean: *Calyx nicaeensis* (Risso, 1827) and *C. tufa* (*Ridley & Dendy, 1886*). The first is the type species of the genus, which is a well-known species characterized by its growing habit (vasiform), blackish color and large size. This species has been widely reported at both the western and eastern Mediterranean in infralittoral and circalittoral bottoms at 3–50 m deep (*Trainito, Baldacconi & Mačić, 2020*). *Calyx* cf. *tufa* clearly differs from *C. nicaeensis* in morphology (massive *vs.* vasiform, respectively), genetics (*COI*: 11 bp difference; *28S*: 43 bp difference) and bathymetry (105–114 *vs.* 3–50 m, respectively). *Calyx tufa* is only known from its type locality at Cape Verde (*Dendy, 1886*) and from the Cantabrian Sea (*Topsent, 1892*). The species that we studied shares many characteristics with *C.* cf. *tufa*, including external morphology, consistency, and skeletal architecture. Unfortunately, the only description available is the one provided by *Dendy (1886)*, which is

too general and matches with the characters of many other *Calyx* spp. (*e.g.*, *Calyx podatypa de Laubenfels 1934*; *Calyx magnoculata* Van Soest, Meesters & Becking, 2014; *Calyx nyaliensis Pulitzer-Finali, 1983*). The large distances between the reports of *C. tufa* and *C.* cf. *tufa*, the strong genetic barriers that separate the two records (the Strait of Gibraltar and the Almeria-Oran front), the generalized low dispersive potential of some sponge species (*Riesgo et al., 2019*; *Griffiths et al., 2021*) and the difference of habitats, are reasons that may suggest that the species that we report here is different from *C. tufa*. Moreover, no intermediate geographical findings have been reported, which would be expected if there was conspecificity (*Topsent, 1928*; *Maldonado, 1992*; *Boury-Esnault, Pansini & Uriz, 1994*; *Sitjà & Maldonado, 2014*; *Sitjà et al., 2019*). It should be noted that *C.* cf. *tufa* is a very large, massive, and easily recognizable sponge, which cannot be easily go unnoticed. However, considering that we did not study the holotype and that no genetic sequences of *C. tufa* are available, the possible conspecificity of *C.* cf. *tufa* with *C. tufa* cannot be completely assessed. Therefore, future work comparing the holotype of *C. tufa* may be needed to determinate if both species are different or conspecific. Considering the mentioned lack of data, here we use the more conservative choice by assigning the present record to *C.* cf. *tufa*.

Order POECILOSCLERIDA *Topsent, 1928*
Family MYXILLIDAE Dendy, 1922
Genus *Melonanchora* Carter, 1874
**Melonanchora emphysema** (*Schmidt, 1875*)
(Fig. 16)

**Synonymised names**
   *Desmacidon emphysema Schmidt, 1875* (genus transfer)

**Material examined**
   CFM-IEOMA-7404/i573, St 31, AM, RD.

**Description**
   Hollow sponge with a detachable, smooth and paper-like ectosome provided with fistulas (Fig. 16A). About 2 cm in diameter. The choanosome is loose and includes sediment. Greyish white in life and after preservation in EtOH.

**Skeleton**
   As in the previous records of the species (Schmidt, 1785; *Vacelet, 1969*; *Pulitzer-Finali, 1983*).

**Spicules**
*Megascleres*
   Tylotes (Figs. 16B and 16C) slightly curved, with roundish ends. Their length tends to be inversely related to their thickness. They measure 359–446–556 × 5–8–11 µm.
*Microscleres*

Spheranchoras (Figs. 16D and 16E) of usual morphology, but uncommon. They measure 36–40–46 × 14–19–23 μm ($n$ = 11).

Arcuate isochela I (Fig. 16F) with well-developed fimbriae and spatulated and bifid alae. They measure 29–42–47 μm.

Arcuate isochela II (Fig. 16G) similar to isochela I, but with rounded alae. They measure 14–18–21 μm.

### Ecological notes

The single specimen was found in AM, on a rhodolith bed between 104 and 138 m deep. It was growing upon a large rhodolith, which was extensively epiphyted by encrusting, massive-encrusting or pedunculated sponges (like *Hamacantha* sp. or *Jaspis* sp.) or pedunculated Axinellids.

### Remarks

The specimen matches well with the previous records of the species, both in external morphology, spicules and skeletal arrangement. This is the third record of this species in the Mediterranean, where it was recorded in the canyon de Cassidaigne in the Gulf of Lions (*Vacelet, 1969*) and Corsica (*Pulitzer-Finali, 1983*). In the North-Atlantic, it has been reported at several localitions: the type locality at Norway (*Schmidt, 1875*), the east Greenland shelf (*Lundbeck, 1905*), the Faroe Plateau (*Hentschel, 1929*) and the north coast of the Iberian Peninsula (*Solórzano, 1990*). The vast distances between the Mediterranean and the North Atlantic reports (being the closest off northern Iberian Peninsula), and the lack of intermediate reports in well-studied areas such as the Alboran Sea, may indicate that Mediterranean and North Atlantic *M. emphysema* are different species, as already discussed by *Vacelet (1969)*.

Order POLYMASTIIDA (Morrow & Cárdenas, 2015)
Family POLYMASTIIDAE (Gray, 1867)
Genus *Polymastia* (Lammarck, 1815)
***Polymastia polytylota* *Vacelet, 1969***
(Fig. 17; Table 9)

### Material examined

CFM-IEOMA-7405/i810, St 39, AM, ROV.

### Description

Rounded sponge, 2 cm high and wide, with two conical papillas (0.5 cm high and 3 mm wide) placed on the upper side of the body (Figs. 17A and 17B). Consistency hard and slightly compressible. Surface smooth to the touch, but microhispid under the stereomicroscope. Cream color before and after preservation in EtOH, with a darker choanosome. The specimen suffered a contraction after collection. *In situ* the sponge was 4.5 cm in height and 4 cm in width, being looser and with its surface full of visible ostia (Fig. 2A).

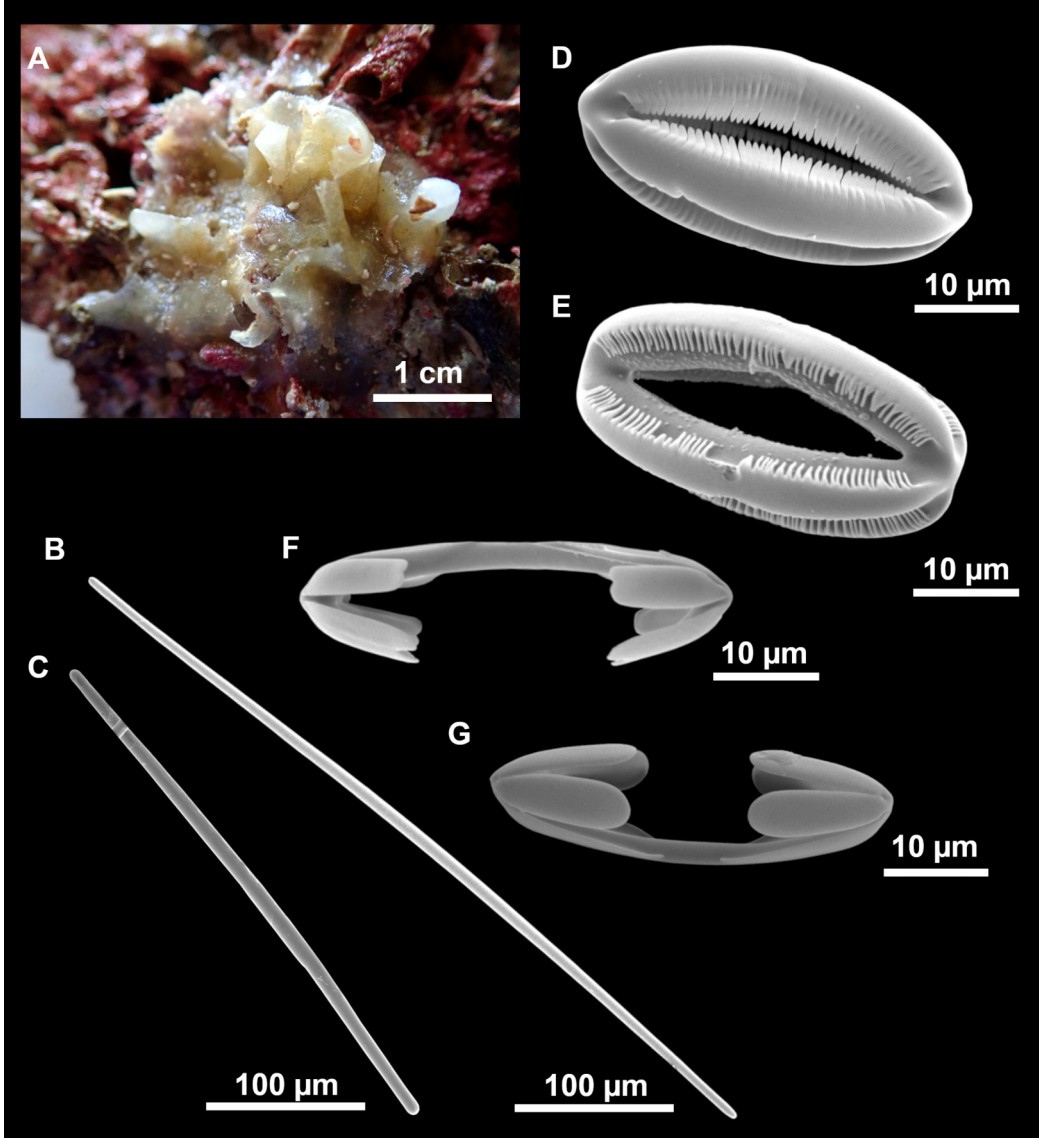

**Figure 16 *Melonanchora emphysema* (*Schmidt, 1875*).** (A) Habitus of CFM-IEOMA-7404/i573 on fresh state, attached to a rodolith. (B–C) Tylotes. (D–E) Spheranchoras. (F) Anchorate isochela I. (G) Anchorate isochela II.

## Skeleton

As in the previous reports of the species (*Vacelet, 1969*; Pulitzer-Finali, 1983; Boury-Esnault, 1987; Boury-Esnault, Pansini & Uriz, 1994).

## Spicules

Principal tylostyles (Figs. 17C and 17C1): straight and fusiform, with several tyles in the proximal half part of the shaft. They measure 438–909–1154 × 8–11–15 µm.

Intermediary tylostyles (Figs. 17D and 17D1): fusiform, with a rounded head, often showing a vesicle. They measure 308–443–586 × 6–7–9 µm.

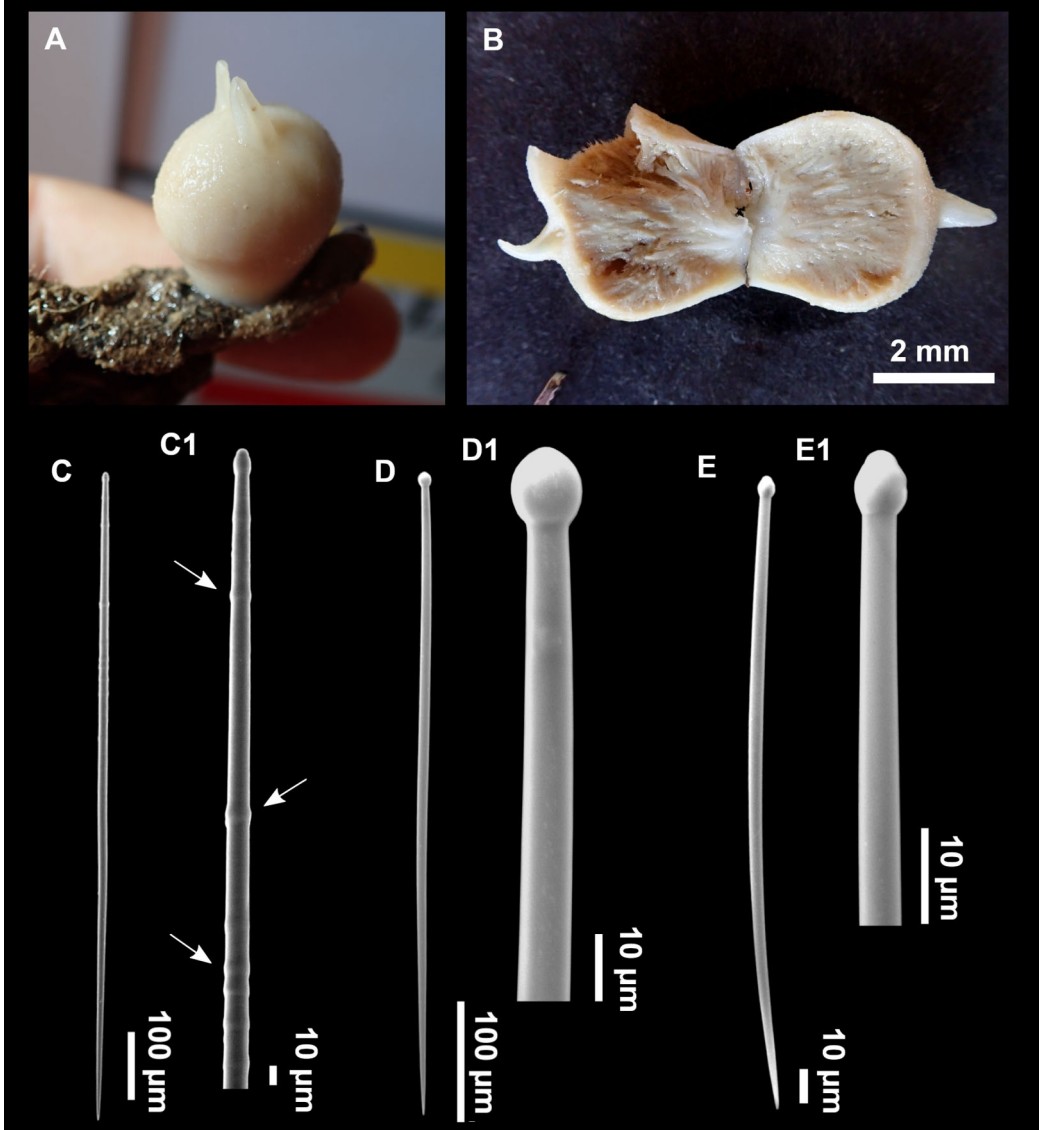

**Figure 17 _Polymastia polytylota_ Vacelet, 1969.** (A–B) Habitus of CFM-IEOMA-7405/i810, on fresh state (A), and preserved in EtOH (B). (C–C1) Principal subtylostyles with detail of the tyles in the shaft (arrows). (D) Intermediary tylostyles with (D1) detail of the head. (E) Ectosomal tylostyles with (E1) Detail of the head.

Ectosomal tylostyles (Figs. 17E and 17E1): slightly curved. They measure 121–166–200 × 2–3–5 µm.

**Ecology notes**

Only one specimen collected in the northern part of the AM, between 352 and 465 m deep, on a rocky bottom characterized by enhanced water movement, with several large _Phakellia_ spp, _Pachastrella_ spp and _Poecillastra compressa_, as well as other _Polymastia_ cf. _polytylota_. Although the present specimen was the only collected, its easy identification

**Table 9 Comparative characters from *Polymastia polytylota* Vacelet, 1969.**

| Specimen | Principal tylostyles | Intermedium tylostyles | Ectosomal tylostyles | Depth | Area |
|---|---|---|---|---|---|
| *Boury-Esnault, (1987)* Redescription of the Holotype | 650–990 × 10–13 | 210–490 × 7–10 | 70–180 × 2–5 | 165–270 | Toulon, but also in Corsica |
| *Boury-Esnault, Pansini & Uriz (1994)* | 668–854–1108 × 5–13–16 | 276–403–509 × 5–11–13 | 94–115–143 × 3–3–4 | Alboran:480 Atl: 362–485 | Alboran Sea and North Atlantic |
| *Pulitzer-Finali (1983)* | 650–810 × 10–13 | 210–490 × 7–10 | 80–120 × 2–3 | 117 | North of Corsica |
| CFM-IEOMA-7405/i810 This work | 438–909–1154 × 8–11–15 | 308–443–586 × 6–7–9 | 121–166–200 × 2–3–5 | 352–465 | AM St 3 |

**Note:**
Depth (m), area (AM, Ausias March) and sampling station (St; see R*study* in Table 1) where these specimens were collected are also shown. Spicule measures are given as minimum-mean-maximum for total length × minimum-mean-maximum for total width. A minimum of 30 spicules per spicule kind are measured, otherwise it is stated. All measurements are expressed in μm. Specimen codes are the reference numbers of the CFM-IEOMA/and author collection.

and the other sightings during ROV transects may suggest that this sponge is quite common in some areas of the Mallorca Channel.

**Remarks**

The present specimen matches with the previous descriptions of the species in external morphology, skeletal architecture and spicule morphometrics. The only difference is that our specimen has two papillae instead of one. The Fig. 2A shows the first *in-situ* image of this species.

This is the first record of this species at the Balearic Islands. In the Mediterranean it is known from the type locality at the Gulf of Lions (*Vacelet, 1969*), the Ligurian Sea (*Vacelet, 1969*; *Pulitzer-Finali, 1983*) and the Alboran Sea, while it has been also reported at the Gulf of Cadiz in the north-eastern Atlantic (*Boury-Esnault, Pansini & Uriz, 1994*).

Genus *Pseudotrachya* Hallmann, 1914
***Pseudotrachya hystrix* (*Topsent, 1890*)**
(Fig. 18; Table 10)

**Material examined**

CFM-IEOMA-7406/i303_A, St 19, AM, RD; CFM-IEOMA-7407/i613, St 32, AM, RD.

**Description**

Roundish and pad-like encrusting sponge, up to 2 cm diameter and 3 mm in height (Fig. 18A). Coloration beige in life and whitish after preservation in EtOH. Very hispid surface. Consistency hard and only slightly compressible. No papillae, oscula and ostia inconspicuous.

**Skeleton** (modified from *Plotkin, Gerasimova & Rapp, 2012*)

Single layered cortex (palisade of microxeas). Main choanosomal skeleton of principal anisoxeas radially arranged, echinating the surface and auxiliary choanosomal skeleton of microxeas (Figs. 18B and 18B1).

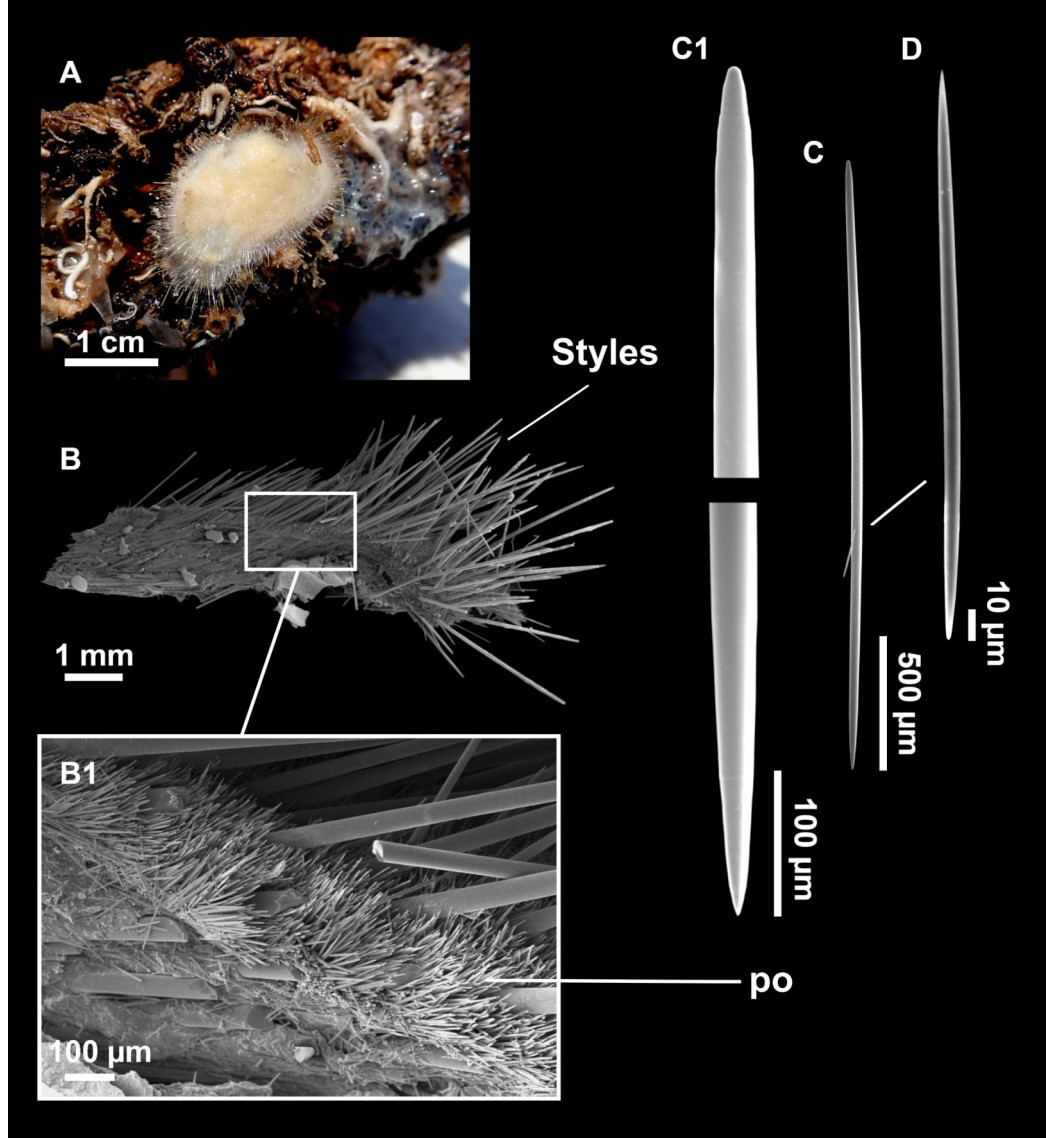

**Figure 18** *Pseudotrachya hystrix* (*Topsent, 1890*). (A) Habitus of CFM-IEOMA-7407/i613 on fresh state. (B–B1) SEM images of the skeletal structure. (po) Palisade of oxeas. (C–C1) Anisoxeas. (D) Microxeas.

**Spicules**

Anisoxeas (Figs. 18C and 18C1): straight and robust, with stepped ends. Intermediary stages between oxeas and styles present. Anisoxea size differs between specimens, measuring 834–1689–3358 × 10–25–42 µm in specimen i303 and 768–2088–3402 × 18–32–45 µm in specimen i613. Small and immature anisoxeas also present, but very scarce, about 500/10 µm.

Microxeas (Fig. 18D): fusiform and measuring 156–185–217 × 4–5–6 µm in specimen i303 and 152–203–270 × 3–5–6 in specimen i613.

**Table 10 Comparative characters from *Pseudotrachya hystrix* (*Topsent, 1890*).**

| Specimen | Anisoxeas | Microxeas | Depth (m) | Area |
|---|---|---|---|---|
| *Topsent (1892)* holotype | up to 7,000 × 70 | 185 × 6 | 318 and 454 | Azores |
| *Topsent (1928)* | nr | nr | 650–914 | Azores |
| Sarà (1959) | 4,000–5,000 × 35–45 | 150–240 × 3–5 | 100 | Tyrrenhian sea |
| *Boury-Esnault, Pansini & Uriz (1994)* | 2,000–3,400–4,300 × 18–44–63 | 200–235–330 × 5–6–7 | 153–568 | Alboran Sea, |
| *Vacelet (1969)* (Various specimens) | St15: 1,000–1,250 × 22–30 St23:>2,000 × 30–35 St34: 1,600–6,600 × 18–40 St46:1,100–4,500 × 20–60 | 110–320 × 3–5 Stylote modifications | St 15: 180 St 23: 210–240 St 34: 270 St 46: 450–550 | St 15: Cassidaigne St 23: Corse St 34: Cassidaigne St 46: Cassidaigne |
| CFM-IEOMA-7406/i303_A This work | 834–1,689–3,358 × 10–25–42 | 156–185–217 × 4–5–6 | 231–302 | AM St 19 |
| CFM-IEOMA-7407/i613 This work | 768–2,088–3,402 × 18–32–45 | 152–203–270 × 3–5–6 | 195–222 | AM St 32 |

Note:
Depth (m), area (AM, Ausias March) and sampling station (St; see R*study* in Table 1) where these specimens were collected are also shown. Spicule measures are given as minimum-mean -maximum for total length × minimum-mean -maximum for total width. A minimum of 30 spicules per spicule kind are measured, otherwise it is stated. All measurements are expressed in μm. Specimen codes are the reference numbers of the CFM-IEOMA/and author collection. nr, not reported.

### Ecology notes

In addition to the two specimens described above, several other *P. hystrix* were collected from rocky slopes of AM and EB, between 195 and 302 m deep, suggesting that this species could be quite common in the Mallorca Channel seamounts. The species is found at rocky slopes, together with other small encrusting sponges such as *Hamacantha* spp., *Bubaris* spp. and the Hexactinellid *Tretodyctium* sp.

### Remarks

This is a well-known species, characterized by their enormous megascleres with unequal tips (oxeote to stylote), and their small microxeas. Variations in the size of megascleres have been previously documented and may be related to ecological factors such as depth, nutrient availability, or temperature (*Maldonado et al., 1999*). However, due to their size, the largest megascleres were mostly broken, which could be a reason for the lack of reports on sizes 5,000–7,000 μm (Table 10).

This is the first record of the species in the Balearic Islands, expanding its geographical distribution in the Mediterranean, where it was previously reported at the Tyrrhenian Sea (*Sarà, 1959*), the Ligurian Sea (*Pulitzer-Finali, 1983*), the Gulf of Lions (*Vacelet, 1969*) and the Alboran Sea (*Boury-Esnault, Pansini & Uriz (1994)*).

Order TETHYIDA Morrow & Cardenas, 2015
Family HEMIASTERELLIDAE Lendenfeld, 1889
Genus *Hemiasterella* Carter, 1879
***Hemiasterella elongata* Topsent, 1928**

### Material examined

CFM-IEOMA-7408/i149_4, St 7, EB, RD; CFM-IEOMA-7409/i337, St 11, EB, BT; CFM-IEOMA-7410/i531, St 24, AM, BT; CFM-IEOMA-7411/POR1066, St 36, south-western Cabrera Archipelago, GOC-73.

### Ecological notes

This species was found at meshophotic bottoms, between 109 and 152 m deep, generally associated with rhodolith beds, or areas with dead rhodoliths on the summits of EB and AM, but also sporadically at the same depths on trawl fishing grounds of the continental shelf of Mallorca.

### Remarks

This is the third record of the species and the third for the Mediterranean, where it was only known from the Alboran Sea (*Sitjà & Maldonado, 2014*). It is also the third report worldwide, considering the type locality at Cabo Verde in the eastern Atlantic (*Topsent, 1928*).

Class HEXACTINELLIDA *Schmidt, 1870*
Subclass HEXASTEROPHORA Schulze, 1886
Order LYSSACINOSIDA Zittel, 1877
Family ROSSELLIDAE Schulze, 1885
Subfamily LANUGINELLIDAE Gray, 1872
Genus *Lanuginella Schmidt, 1870*
**Lanuginella pupa** *Schmidt, 1870*
(Fig. 19; Table 11)

### Material examined

CFM-IEOMA-7412/i286_1, CFM-IEOMA-7413/i286_2 and CFM-IEOMA-7414/i286_3, St 18, AM, RD.

### Description

Tubular (CFM-IEOMA-7412/i286_1) to calyx-like (CFM-IEOMA-7413/i286_2 and CFM-IEOMA-7414/i286_3) sponges (Fig. 19A), up to 4 cm high and 2 cm in diameter. Surface smooth, but slightly hispid at localized areas. Fragile consistency and soft touch. Dirty white color in life and white after preservation in EtOH. All the three specimens present a single, circular oscule at the upper part of the body. One of the calyx-like specimens (CFM-IEOMA-7413/i286_2) has a minute and short peduncle.

### Skeleton

As usual for the species (see *Ijima, 1904*; *Tabachnick, 2002* and *Sitjà et al., 2019* for detailed descriptions)

### Spicules

Choanosomal diactines (Fig. 19B): long and slim, slightly sinuous, with four vestigial tubercles in the center (Fig. 19B1), which may have swelings all over the shaft and spines on their tips. They measure 245–1611 × 3–15 μm.
Choanosomal hexactines (Fig. 19C): with actines of different lengths, sometimes sinuous. They measure 349–983 × 10–25 μm ($n = 12$).
Hypodermal pentactines (Fig. 19D): with a ray reduced to a stump or absent. Proximal ray much larger than the others and perpendicularly arranded. Rays are smooth or slightly

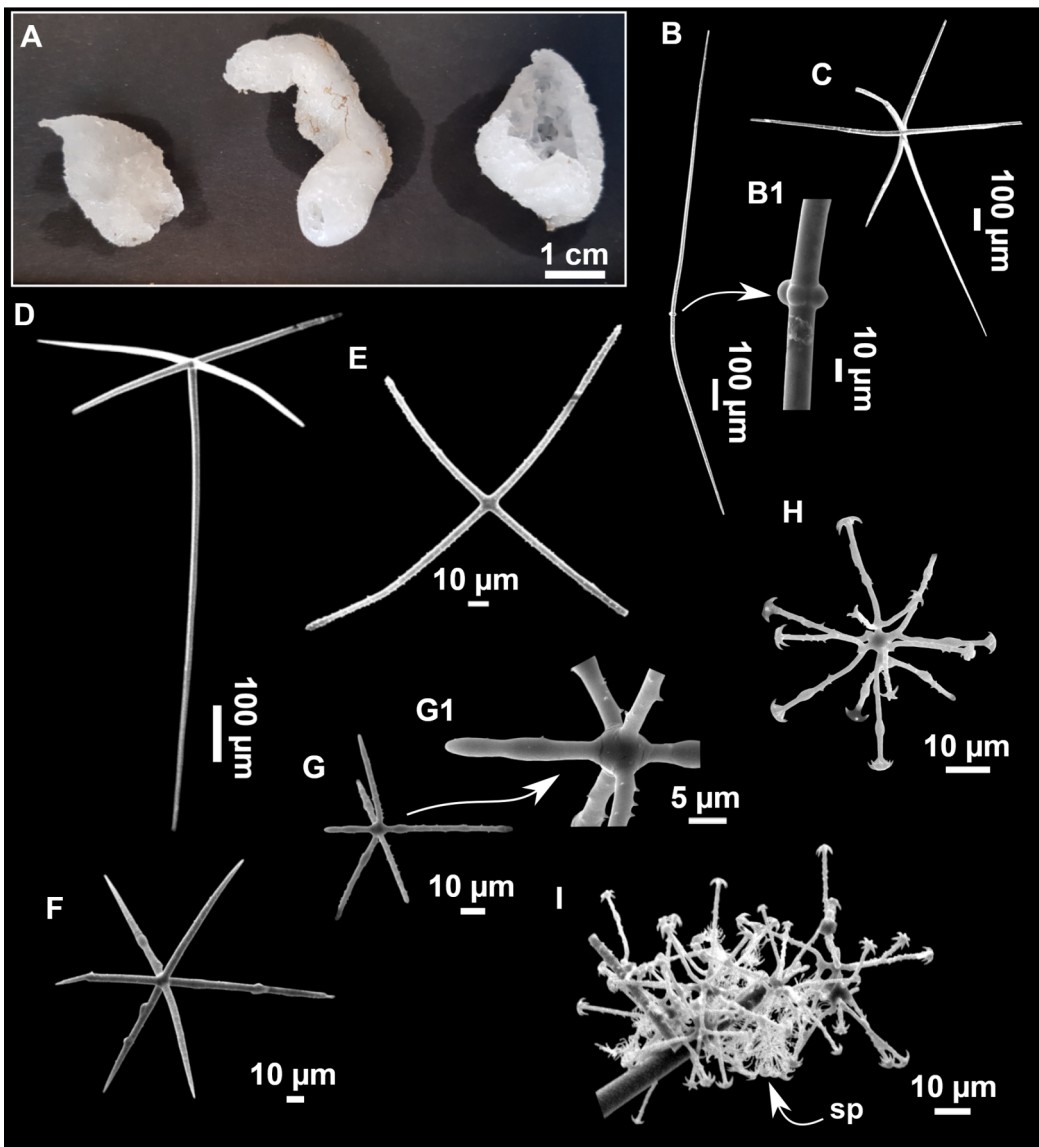

**Figure 19 *Lanuginella pupa Schmidt, 1870*.** (A) Habitus of CFM-IEOMA-7413/i286_2 (left), CFM-IEOMA-7412/i286_1 (middle), i286_3/CFM-IEOMA-7414 (right) preserved in EtOH. (B–I) SEM images of spicules from CFM-IEOMA-7412/i286_1. (B) Choanosomal diactine with (B1) detail of the four central tubercles. (C) Choanosomal hexactine. (D) Hypodermal pentactine. (E) Stauractine. (F–G) Atriala hexactines with (G1) Detail of the spines of (G). (H) Discohexaster. (I) Agglomeration of discohexasters, with a strombiloplumicome (sp) beneath.

rugose. Proximal ray measuring 242–950 × 7–19 μm (*n* = 8) and perpendicular rays measuring 137–850 × 4–20 μm (*n* = 28).

Stauractines (Fig. 19E): with four actines perpendicularly arranged one another in the same plane. They are straight or slightly curved, strongly spined, with roundish tips. They measure 61–111 × 3–5 μm (*n* = 23).

Dermal hexactines (not shown): uncommon. Rugose, with the proximal ray slightly longer than the distal one. Overall measures: proximal rays 151 × 7 μm (*n* = 1), distal rays 105 × 6 μm (*n* = 1) and perpendicular rays 68–110 × 2–6 μm (*n* = 3).

**Table 11 Comparative characters from *Lanuginella pupa* Schmidt, 1870.**

| Specimen | Parenchimalia | | Dermalia | | Gastralia | Hexasters | | Depth | Area |
|---|---|---|---|---|---|---|---|---|---|
| | Choanosomal Diactine | Choanosomal Hexactine | Hypodermal Pentactine | Other | Atrial hexactine | Discohexaster | Strombiplumicomes | | |
| *Schmidt (1870)* holotype | nr | | | | | | | | Cape Verde (Atlantic Ocean) |
| *Schulze (1897)* | nr | nr | nr | Stauractines: 160–200 | nr | 32–80–100 | 40 | 201 | Little Ki Island (Banda Sea) |
| Ijima (1904) Several specimens | up to 4,000 × 22 | variable in size, up to 2,000 × 30 (as oxyhexactine) | 1,000 × 34 | Stauractines rarely tauactines, 220–330 (average length) × 7 | 220–330 × 7 | 40–90 | 34–76 | 183–572 | Dōketsba, Okinose, Mochiyama (Japan), Vries Island (Vries Strait, Pacific Ocean) |
| Okada, 1932 | 3,500–5,000 × 100 | 3,000 × 60 | nr | nr | nr | 45–80 | 50 | 180 | Kagoshima Gulf (Eastern China Sea) |
| Burton, 1956 | nr | | | | | | | | |
| *Sitjà & Maldonado (2014)* | 325–3,000 × 4–7 | 250–850 × 6–13 | Perp: 170–850 × 4–10 Prox: 242–950 × 8–12 | Abundant stauractines, scarce pentactins, tauactins and hexactins 43–140 × 2–6 | 46–150 × 2–6 | 30–70 | nf | 690 | Gulf of Cadiz (Mud volcano, North Atlantic Ocean) |
| CFM-IEOMA-7412/i286_1 This work | 586–1,900 × 7–14 (n = 6) | 664–983 × 10–25 (n = 3) | Perp: 175–343–435 × 6–13–20 (n = 13) Prox:323–777 × 7–19 (n = 3) | Stauractins: 61–91 × 3–5 (n = 5) Hexactins: Perp: 101 × 6 (n = 1) Prox: 151 × 7 (n = 1) Dist: 105 × 6 (n = 1) Paratetractins: 77–4 (n = 1) | Prox: 108–113 × 5–6 (n = 2) Dist: 70–95 × 4–7 (n = 2) Perp: 71–150 × 4–7 (n = 9) | 43–57–70 (n = 12) | 20–38 (n = 5) | 220–275 | AM St 18 |
| CFM-IEOMA-7413/i286_2 This work | 245–1,726–2,586 × 3–9–12 (n = 19) | 492–920 × 16–20 (n = 4) | Perp: 242–519 × 8–15 (n = 7) | Stauractins: 79–132 × 3–5 (n = 5) Hexactins: Perp: 68–110 × 2–5 (n = 2) | Prox: 107 × 4 (n = 1) Dist: 97 × 5 (n = 1) Perp: 83–145 × 3–6 (n = 7) | 49–76 (n = 7) | nf | 220–275 | AM St 18 |
| CFM-IEOMA-7414/i286_3 This work | 528–1,533–2,611 × 3–9–15 | 349–926 × 10–19 (n = 5) | Perp: 137–437 × 9–17 (n = 8) Prox: 516–831 × 11–14 (n = 5) | Stauractins: 67–91–111 × 3–4–5 (n = 13) Hexactins: n.f. | Prox: 119–159 × 4–6 (n = 5) Dist: 79–102 × 4–6 (n = 5) Perp: 70–96 × 4–6 (n = 5) | 45–53–62 (n = 10) | 20 (n = 1) | 220–275 | AM St 18 |

**Note:**

Depth (m), area (AM, Ausias March) and sampling station (St: see R*study* in Table 1) where these specimens were collected are also shown. Spicule measures are given as minimum–mean–maximum for total length × minimum–mean–maximum for total width. A minimum of 30 spicules per spicule kind are measured, otherwise it is stated. All measurements are expressed in μm. Specimen codes are the reference numbers of the CFM-IEOMA/and author collection. nr, not reported; nf, not found.

Paratetractin: only a single spicule observed, measuring77/4 ($n$ = 1).

Atrialia hexactines (Figs. 19F and 19G): common. Slightly rough to smooth. Overall measures: proximal rays 107–159 × 4–6 μm ($n$ = 8), distal rays 70–102 × 4–7 μm ($n$ = 8) and perpendicular rays 70–150 × 3–7 μm ($n$ = 22).

Discohexasters (Fig. 19H): rather uncommon. Some with underdeveloped, twisted rays. They measure 43–76 μm ($n$ = 29).

Strobiloplumicomes (Fig. 19I): very rare and not found in specimen CFM-IEOMA-7413/i286_2. They measure: 20–38 μm ($n$ = 6).

### Ecological notes

Species found only at one station located in a rocky slope at SO, between 220 and 275 m deep. It was associated with fossil ostreid reefs and carbonated rocks, together with other encrusting sponges like *Hamacantha* sp., *Bubaris* sp., and *Jaspis* sp.

### Remarks

This poorly-known species is the single representative of the genus *Lanuginella*, reported at several distant locations around the world: Kagoshima Gulf at the Sea of China (*Okada, 1932*), Ki Island at the Sea of Banda (*Schulze, 1897*). In the northern Atlantic it was recorded at Cabo Verde (*Schmidt, 1870*), the Gulf of Cadiz (*Sitjà et al., 2019*) and the Strait of Gibraltar (*Topsent, 1895*). This is the first record of the species in the Mediterranean Sea, increasing its already wide distribution. However, a revision of the species is needed, and it is likely that such a cosmopolitan distribution may indicate that *L. pupa* represents a species complex. However, deep-sea species tend to be more widely distributed than shallow ones, probably because of the uniformity of in the environmental conditions (*McClain & Hardy, 2010*). A detailed examination of worldwide specimens, combined with molecular methods, may shed more light on it.

## DISCUSSION

### Biogeography and seamount singularity

The present study increases the knowledge of the sponge diversity of the Mediterranean seamounts. We describe a new genus, three new species, and 15 new geographical reports, including two new reports for the Mediterranean Sea. This study also highlights *Foraminospongia balearica* **sp. nov.** as one of the most common sponges at AM and EB, being large and easily distinguishable. This species was never recorded at other previously explored Mediterranean seamounts or ridges of a similar depth range, such as the Seco de los Olivos or the Alboran Ridge, whose sponge fauna has been already studied (*Sitjà & Maldonado, 2014*; *Würtz & Rovere, 2015*; *De la Torriente et al., 2018*). Therefore, the Mallorca Channel seamounts may be considered unique faunal refuges, appealing to what is called the "Seamount endemism hypothesis" (*De Forges, Koslow & Poore, 2000*), which suggests that geographical separation of seamounts is reflected by genetic isolation of their fauna, which promotes speciation by vicariance. This hypothesis has been questioned, as some works have shown that benthic fauna (including sponges) is well connected among isolated seamounts (*Samadi et al., 2006*; *Ekins et al., 2016*).

However, others have shown structured populations between seamounts (*Castelin et al., 2010*), or between seamounts and the continental shore populations (*Crochelet et al., 2020*). Other authors suggest that there is a mixture of panmictic and structured populations, largely dependent on the characteristics of the single species nature (*Rogers, 2018*). If we consider that the dispersal of sponges tend to be very limited (*Maldonado, 2006*; *Riesgo et al., 2019*; *Shaffer et al., 2020*; *Griffiths et al., 2021*), it is plausible that certain seamount sponge populations are highly structured. This limitation in the dispersal may be enhanced in isolated seamounts or in those with peculiar or unique ecological characteristics.

In this sense, both AM and EB have very shallow summits and are placed in an area of special oligotrophy (*e.g.*, *Bosc, Bricaud & Antoine, 2004*; *Uitz et al., 2012*). The nearest habitat with similar features is the continental shelf of the Balearic promontory, although these areas tend to be under the impact of bottom-trawling (*Farriols et al., 2017*; *Ordines et al., 2017*), with the consequent impoverishment of benthic communities (*Jennings & Kaiser, 1998*). In fact, most of the species of SO, AM and the EB had not been found at the continental shelf of the Balearic Islands (*Bibiloni, 1990*; *1993*; *Grinyó et al., 2018*; *Santín et al., 2018*), except for *Phakellia robusta*, *P. hirondellei* (*Santín et al., 2018*), *Petrosia* (*Petrosia*) *raphida*, and *Hemiasterella elongata* (this work).

The particular conditions of the Balearic Islands, extreme oligotrophy, geographical isolation, low fishing pressure and heterogeneity of habitats (*Quetglas et al., 2012*; *Massutí et al., 2014*) suggest this area is a hotspot of sponge diversity, with much of its fauna still unknown, especially at depth below 90 m (*Bibiloni, 1990*; *Santín et al., 2018*; *Díaz et al., 2020*). In recent years, this high diversity has been evidenced by the presence of rich benthic assemblages (*Ordines & Massutí, 2009*; *Barberá et al., 2012*; *Ordines et al., 2011*), as well as by a high number of new species and new geographical reports (*e.g.*, *Kovačić, Ordines & Schliewen, 2017*; *Kovacic et al., 2019*; *Guzzetti et al., 2019*; *Ordines et al., 2019*; *Díaz et al., 2020*). Thus, there is a need to find out which sponge species inhabit those waters and how much do they contribute to the benthic biomass. Sponges are key components of the benthic ecosystems, playing important biogeochemical roles (*De Goeij et al., 2013*) and serving as food or refuge to many other animals (*Maldonado et al., 2016*). Future works should characterize those benthic habitats of the continental shelf and slope around the Balearic Islands that are potentially similar to those of the Mallorca Channel seamounts (*e.g.*, non-impacted sedimentary and rocky bottoms with rhodoliths and gravels located between 90 and 150 m deep and rocky slopes down to 400–500 m deep). Then, both biocenosis should be compared to confirm the singularity of the habitats of the Mallorca Channel seamounts.

## Integrative taxonomy

The generalized lack of distinctive characters has caused sponges to be one of the most difficult groups to classify. This difficulty is also reflected by sponge phylogenetic relationships, with polyphyletic taxa present in all the levels of the Linnean classification (*e.g.*, *Cárdenas, Perez & Boury-Esnault, 2012*; *Díaz et al., 2020*). Thus, the use of both morphology and molecular markers is central to the improvement of the knowledge of this group of organisms. Following this approach, here we have proposed the new genus

*Foraminospongia* to be erected in the family Hymerhabdiidae, supported by the two new species *Foraminospongia balearica* sp. nov. and *Foraminospongia minuta* sp. nov., confirmed by morphological traits and both *COI* and *28S* markers. On the other hand, the species *Heteroxya* cf. *beauforti* has shown no variability in its *COI* sequence relative to its North Atlantic congenerics, which highlights the importance of morphology and the need to combine both approaches. The *COI* is known to be a low-resolution marker to discriminate species of sponges (*Wörheide 2006*), so we also sequenced the more variable *28S* marker. However, no *28S* sequences are currently available in any database for comparison with the other *Heteroxya* spp.: this issue should thus be adressed in the future.

A key subject in sponge taxonomy is the robustness of the skeletal characters as a species diagnostic tool, and how reliable they are for discriminating species and populations. Reliable discrimination is further complicated by the fact that skeletal elements may change depending on environmental conditions such as temperature, depth, or nutrient concentration; skeletal elements may also change due to intraspecific plasticity, overall modifying length, width, morphology, and even their presence or absence (*Cárdenas, Perez & Boury-Esnault, 2012*; *Abdul Wahab et al., 2020*). No consensus has ever been reached to consider a given morphological deviation as enough evidence to erect a new species, a fact that remains arbitrary. We have found differences in the spicular morphometry between the specimens of the Balearic Islands and specimens of other areas of the Mediterranean and the North Atlantic Ocean; these differences have been described here for most of the species to some extent. Since the dispersive potential, long-distance connectivity, and speciation of sponges are poorly understood, most of the diagnosis in the present work were performed under a conservative approach, only proposing new species when we found solid morphological evidence. Taking this into account, factors like vast geographical distances, presence of oceanographic barriers or minor morphometric differences were not considered enough evidence for species delimitation. In the case of *Calyx* cf. *tufa*, its potential conspecificity with the North Atlantic species *C. tufa* cannot be discarted. We did not get access to any material of *C. tufa*, and no sequences are available for comparison; moreover, the original description is too vague and general. However, as stated above, the absence of any intermediate records of such a big, conspicuous, and easily recognizable sponge is noreworthy. Also, the recorded depths of *C. tufa* for the Atlantic are much deeper than those for *C.* cf. *tufa* (219 and 300 m *vs.* 105–114 m). Future work is need to clarify if both species are synonyms, or if *C.* cf. *tufa* is a new species for science.

## ACKNOWLEDGEMENTS

The authors wish to thank the captain and crew of R/Vs *Ángeles Alvariño*, *Sarmiento de Gamboa* and *Miguel Oliver*, as well as the participants in the INTEMARES surveys at the Mallorca Channel seamounts and the MEDITS 2017, 2019 and 2020 surveys. Special thanks to Dr. Ferran Hierro (University of the Balearic Islands) for the technical assistance at the scanning electron microscope, to Dr. Paco Cárdenas, for taxonomic advice, to Maria Teresa Farriols and Helena Marco for the logistic work during the development of the INTEMARES surveys. Lastly, we would like to thank Professor Maurizio Pansini and the

two anonymous reviewers for their constructive comments and corrections to the original manuscript.

### Funding

This research has been performed within the scope of the LIFE IP INTEMARES project, coordinated by the Biodiversity Foundation of the Ministry for the Ecological Transition and the Demographic Challenge. It receives financial support from the European Union's LIFE programme (LIFE15 IPE ES 012). The MEDITS surveys are co-funded by the European Union through the European Maritime and Fisheries Fund (EMFF) within the National Program of collection, management and use of data in the fisheries sector and support for scientific advice regarding the Common Fisheries Policy. Julio A. Diaz is supported by FPI contract co-funded by the Regional Government of the Balearic Islands and the European Social Fund. Sergio Ramírez-Amaro is funded by a post-doctoral contract from the regional government of the Balearic Islands, co-funded by the Regional Government of the Balearic Islands European Social Fund. The funders had no role in study design, data collection and analysis, decision to publish, or preparation of the manuscript.

### Grant Disclosures

The following grant information was disclosed by the authors:
LIFE IP INTEMARES.
Biodiversity Foundation of the Ministry for the Ecological Transition and the Demographic Challenge.
European Union's LIFE Programme: LIFE15 IPE ES 012.
MEDITS.
European Union through the European Maritime and Fisheries Fund (EMFF).
National Program.
Common Fisheries Policy.
FPI.
Regional Government of the Balearic Islands and the European Social Fund.
Government of the Balearic Islands.

### Competing Interests

The authors declare that they have no competing interests.

### Author Contributions

- Julio A. Díaz conceived and designed the experiments, performed the experiments, analyzed the data, prepared figures and/or tables, authored or reviewed drafts of the paper, and approved the final draft.
- Sergio Ramírez-Amaro conceived and designed the experiments, performed the experiments, analyzed the data, authored or reviewed drafts of the paper, and approved the final draft.

- Francesc Ordines conceived and designed the experiments, performed the experiments, analyzed the data, authored or reviewed drafts of the paper, and approved the final draft.

## Data Availability

The *COI* (MW858346–MW858351 and MZ570433) and *28S* (MW881149–MW881153) sequences are available at GenBank.

All the specimens were deposited in the Marine Fauna Collection (http://www.ma.ieo.es/cfm/) based at the Centro Oceanográfico de Málaga (Instituto Español de Oceanografía), with the numbers from CFM7356 to CFM7417 (Table S1).

## New Species Registration

The following information was supplied regarding the registration of a newly described species:

Publication:

urn:lsid:zoobank.org:pub:47EC2384-A88C-4654-8425-A7A46BC47AC5

Foraminospongia new **gen. nov.**:

urn:lsid:zoobank.org:act:5CC3710C-0749-47BD-B2BA-92158E5B0758

Foraminospongia balearica **sp. nov.**:

urn:lsid:zoobank.org:act:A1582919-710A-4E64-BBB5-0908B81F5206

Foraminospongia minuta **sp. nov.**:

urn:lsid:zoobank.org:act:8EE74D72-BFF2-4FC4-8E43-354FCC35F11F

Heteroxya cf beauforti

urn:lsid:zoobank.org:act:BF907E49-598F-4348-87F5-AEA11F0D3944

Paratimea massutii **sp. nov.**:

urn:lsid:zoobank.org:act:5A9B89F8-2777-4AB7-845F-9EC4BB91D518.

## Supplemental Information

Supplemental information for this article can be found online at http://dx.doi.org/10.7717/peerj.11879#supplemental-information.

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
