# Peer review of "Sponges of Western Mediterranean seamounts: new genera, new species and new records"

_PeerJ, doi:10.7717/peerj.11879_

## Round 0.1 · original submission · Major Revisions

Three reviewers have highlighted the strengths of you submitted manuscript and have proposed a number of changes to improve the quality. Please consider also the annotated manuscript files provided by the three reviewers. The proposed changes are mandatory for a final acceptance of your manuscript.

I look forward to your revised manuscript.

Reviewer 1 ·

Basic reporting

There are a number of spelling and grammatical errors which can be confusing. I have provided suggestions for improvement in my review but encourage the authors to seek assistance from a native English speaker to review the article prior to re-submission. Technically, the manuscript is sound and comprehensive.

There are a few missing references from the reference list. I suggest the authors carefully cross-check between in-text and the reference list. Otherwise, the references are sufficient in providing the necessary background information.

Figures and tables are well presented.

Experimental design

The study is taxonomy/ biodiversity focused and descriptive. The research question is well defined and technical approach sound.

Validity of the findings

The findings are novel and important towards building fundamental understanding of sponge diversity on seamounts in the Mediterranean.

Additional comments

A lot of work has gone into this study, to identify the diversity and also to describe new species of sponges that are found on seamounts. Technically, I find little fault with the study, however, I do encourage the authors to thoroughly review and correct any critical grammatical errors contained. I have provided comments as annotation to the pdf copy of the manuscript for careful consideration.

Annotated reviews are not available for download in order to protect the identity of reviewers who chose to remain anonymous.

·

Basic reporting

The paper reports the original results of important and extensive surveys performed on the seamounts off the Balearic Islands which are of remarkable importance. The study of the seamounts is particularly interesting for a better knowledge of the mesophotic zone sponge fauna. There is still a gap of knowledge of deep habitats and the present research contributes to fill it. The involved ecological and biogeographical aspects have been considered.
Authors are certainly more familiar with genetics and phylogeny than with sponge taxonomy, biogeography and so on. They do not always use the current terminology about sponge morphology, spicules, skeleton and character description. Several publications as the thesaurus of sponge morphology or the sponge guides may help them and I invite to consult them during the MS revision. I have corrected the manuscript – spending a lot of time and with a considerable effort – and all my notes and comments are reported on the pdf file. The English language requires a careful revision. I have made several corrections on the MS where the meaning was unclear or mistakes were evident, but this is not my task nor am I a mother tongue reviewer.
The introduction is complete and detailed and the structure of the paper conforms to the rules of the Journal.
Figures are exhaustive and of good quality. The notes on the MS help to make clearer the legends both of figures and tables.
The number of tables in my opinion is excessive (especially as to those regarding single species) but this depends on the policy of the Journal about the space to allow for each submitted paper. As to the position of the tables I would prefer to have them placed within the text and not at the end of the MS but this one also is and editorial choice.
References are adequate and correct. The insertion of a few more citations is suggested with notes on the manuscript.

Experimental design

Sampling was correctly performed and specimens were studied with modern techniques and adequate facilities which are well described in the method paragraph. Sponge taxonomy is the main topic of the study but phylogenetic aspects are treated in detail.

Validity of the findings

The erection of new species is justified and they are correctly described taking into account the genetic aspects. Probably also the choice of attributing two of them to a new genus is correct. The skeletal characters of the new genus are clearly described but the images illustrating them are not fully convincing. If possible they should be replaced with better ones or integrated with a drawing.
The choice of the name Pseudoaplysina for the new genus on the basis of a mere morphological and far from univocal likeness - in my opinion - has to be rejected. The genus Aplysina has nothing to do with the new one and the use of this name is misleading.
I have no suggestions about the choice of the new name but I recommend being cautious also when using the term pseudo. This should be a motivated choice suggesting the idea of “false” in relation to a valid term of comparison.

Additional comments

I invite you to read carefully the pdf file reporting corrections and suggestions and to follow it during the revision of the MS

Reviewer 3 ·

Basic reporting

In line with yours standards

Experimental design

Very good

Validity of the findings

Very good

Additional comments

Dear Editor and authors,
the manuscript titled "Sponges of Western Mediterranean seamounts: new genera, new species and new records" is a beautiful work, well written (although I am not a native speaker). The manuscript describes the biodiversity of sponges of some seamounts of the Mediterranean Sea with the description of a new genus and new sponges. Despite the scientific value of this work, I would like to propose some points that I do not agree with the authors (in the word of the manuscript that I have attached there are the details).
My opinion is that this MS can be accepted after medium review.
1. Abstract “Background”, Authors should write more than just “are poorly explorer”!
2. “Results”, Some data are not true, for example distribution of P. vansoesti (see word MS file);
3. INDEX: In my opinion it makes no sense!
4. Introduction. Some sentences are followed by quotations that are now a bit dated! The names of the species that are inserted should always be checked and updated according to the systematic. Whenever a species is cited for the first time, the authors and the year must be entered (not in italics);
5. Materials and methods. Ok, just a few minor changes;
6. Results. Errors in the subgenera, lack of the authors of the species;
7. Systematic. The name “Pseudoaplysina” for the new genus, in my opinion is that this name is beautiful but very misleading, having nothing to do with the genus Aplysina, the external morphology cannot be used in this sense! However, if you zoom in on the photo, the surface is completely different from that of Aplysina. In my opinion, the name of the genus should be changed. Very good descriptions and rest; the description of the skeleton would go before the spicules; I would divide "Taxonomic remarks" and "Molecular remarks". We come to the description of the well-known species like some Phakellia species (see text), you have done a good job, but the descriptions of these species are already well done in other works and the images of the spicules, in my opinion it is an unnecessary repetition, I would leave only the part of the distribution and ecology.
Heteroxya mediterranea sp. nov. I'm sorry but this species cannot be considered valid, the authors themselves write " the COI sequence is identical to two others …", if no other molecular evidence is found, perhaps with other markers, this species must be eliminated from the manuscript.
Rhabdobaris implicata, Haliclona (Soestella) fimbriata, Petrosia (Strongylophora) vansoesti, Petrosia (Petrosia) raphida and Hemiastrella elongata other species known and already described very well; in my opinion descriptions are superfluous.
8. Discussion. Everything that I have marked in red must be rewritten for me. It is absolutely not true that classical taxonomy should be replaced with genetics, the latter has made huge mistakes over the years! They should be used together, but it's still nice and above all scientifically correct to be able to create new species without genetics!
9. Figures and tables: Ok, apart from the little things marked in the MS file.

Annotated reviews are not available for download in order to protect the identity of reviewers who chose to remain anonymous.

---

## Round 0.2 · Minor Revisions

We are almost there. As identified by reviewer 3, a few minor changes have to be made before a final acceptance of your manuscript.

Reviewer 3 ·

Basic reporting

already commented in the first revision

Experimental design

already commented in the first revision

Validity of the findings

already commented in the first revision

Additional comments

already commented in the first revision

Annotated reviews are not available for download in order to protect the identity of reviewers who chose to remain anonymous.

---

## Round 0.3 · accepted · Accept

Thank you very much for the thorough revision of your manuscript and the consideration of the reviewer's proposals. I agree with the definition of the Western Mediterranean Sea in your manuscript.